# ITERRESEARCH: RETHINKING LONG-HORIZON AGENTS WITH INTERACTION SCALING

**Guoxin Chen**[1,4], **Zile Qiao**[2,*], **Xuanzhong Chen**[2], **Donglei Yu**[2], **Haotian Xu**[3]
**Wayne Xin Zhao**[1,4,*], **Ruihua Song**[1,*], **Wenbiao Yin**[2], **Huifeng Yin**[2], **Liwen Zhang**[2]
**Kuan Li**[2], **Minpeng Liao**[2], **Yong Jiang**[2], **Pengjun Xie**[2], **Fei Huang**[2], **Jingren Zhou**[2]
[1]Gaoling School of Artificial Intelligence, Renmin University of China
[2]Tongyi Lab, Alibaba Group, [3]OpenRLHF
[4]Beijing Key Laboratory of Research on Large Models and Intelligent Governance
{gx.chen.chn, batmanfly}@gmail.com, songruihua_bloon@outlook.com
{qiaozile.qzl, yongjiang.jy}@alibaba-inc.com

## ABSTRACT

Recent advances in deep-research agents have shown promise for autonomous knowledge construction through dynamic reasoning over external sources. However, existing approaches rely on a mono-contextual paradigm that accumulates all information in a single, expanding context window, leading to context suffocation and noise contamination that limit their effectiveness on long-horizon tasks. We introduce **IterResearch**, a novel iterative deep-research paradigm that revisits long-horizon research through the lens of Interaction Scaling. Instead of relying on linear context accumulation, we adopt an MDP-inspired architecture with strategic workspace reconstruction. By maintaining an evolving report as memory and periodically synthesizing insights, our approach preserves consistent reasoning capacity across arbitrary exploration depths. To effectively train this paradigm, we employ Efficiency-Aware Policy Optimization (EAPO), a training strategy that adapts geometric reward discounting to incentivize efficient exploration and utilizes adaptive downsampling for stable distributed training. Extensive experiments demonstrate that IterResearch achieves substantial improvements over existing open-source agents with average +14.5pp across six benchmarks and narrows the gap with frontier proprietary systems. Remarkably, our paradigm exhibits unprecedented interaction scaling, extending to 2048 interactions with dramatic performance gains (from 3.5% to 42.5%), and serves as an effective prompting strategy, improving frontier models by up to 19.2pp over ReAct on long-horizon tasks. These findings position IterResearch as a versatile solution for long-horizon reasoning, effective both as a trained agent and as a prompting paradigm for frontier models.

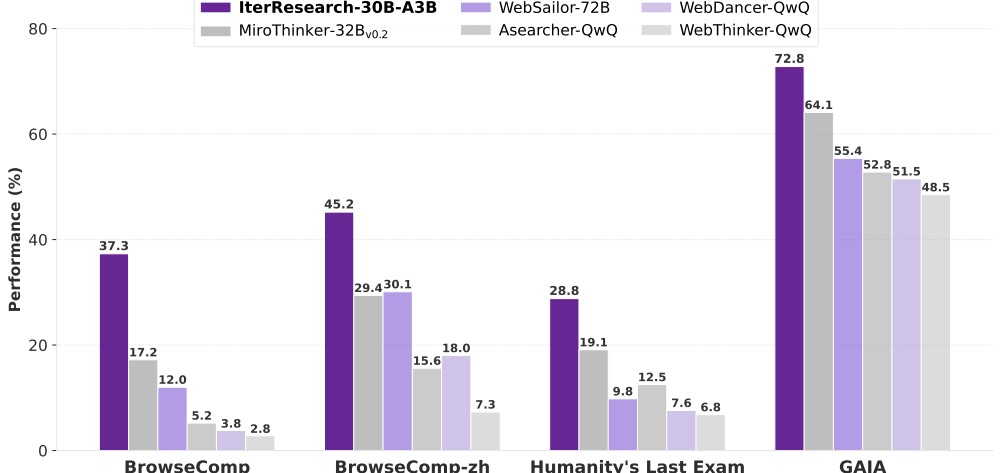

Figure 1: Performance of IterResearch against state-of-the-art open-source long-horizon agents.

---

*Corresponding Authors.  ○ Code

# 1 INTRODUCTION

Recent advances in deep-research agents represent a transformative shift for Large Language Models (LLMs), moving beyond passive knowledge acquisition from the model itself towards autonomous agents that construct knowledge through dynamic reasoning over external sources (OpenAI, 2025a; Google, 2025; xAI, 2025; Perplexity, 2025; Anthropic, 2025; Team et al., 2025). These frontier proprietary systems have demonstrated remarkable performance on long-horizon tasks that require sustained reasoning and information-seeking capabilities over extended interactions.

When tackling long-horizon tasks, recent works (Chen et al., 2025a; Song et al., 2025; Zheng et al., 2025b; Jin et al., 2025; Li et al., 2025c;a; Tao et al., 2025) typically append all retrieved information and intermediate reasoning steps to a single, continuously expanding context window, which we term the *mono-contextual paradigm*. While straightforward to implement, this paradigm fundamentally undermines the sustained reasoning capabilities required for long-horizon tasks: (1) **context suffocation**: as the context window fills with all prior interactions, the available space for model reasoning progressively shrinks, forcing increasingly constrained responses that ultimately degrade into premature or superficial conclusions. (2) **noise contamination**: irrelevant information from web searches and early exploration errors become permanently embedded in the context, creating cascading interference that dilutes signal quality throughout the entire reasoning process.

To address these limitations, we introduce **IterResearch**, a novel *Iterative Deep-Research Paradigm* that fundamentally reimagines how autonomous agents maintain sustained reasoning capacity in long-horizon scenarios. Our key insight is that effective long-horizon research requires periodic synthesis and strategic forgetting—capabilities absent in current mono-contextual approaches. Specifically, we adopt an MDP-inspired architecture to structure deep research with a distinctive state design: rather than maintaining an ever-expanding history, each state is a strategically reconstructed workspace containing only essential elements: the question, an evolving report serving as the agent's memory, and the immediate context needed for current reasoning. This iterative structure, where future exploration depends only on the current reconstructed state rather than the entire history, enables the agent to maintain consistent reasoning capacity across arbitrary exploration depths while naturally circumventing the degradation that plagues mono-contextual approaches.

To effectively train this iterative paradigm, we employ Efficiency-Aware Policy Optimization (EAPO), a training strategy specifically designed for IterResearch. EAPO addresses two critical challenges unique to our iterative paradigm: First, recognizing that not all successful trajectories are equally valuable, we adapt standard geometric discounting to create efficiency-aware rewards that geometrically discount based on trajectory length—agents reaching correct conclusions through concise, focused exploration receive higher rewards than those requiring extensive iterations. Second, since our paradigm naturally decomposes trajectories into independent training samples at each round, we employ *adaptive downsampling* to handle the variable sample counts based on data-parallel size, ensuring stable distributed training while preserving over 99% of training data.

Extensive experiments demonstrate that IterResearch significantly outperforms existing open-source agents, achieving an average improvement of 14.5 percentage points (pp) across six challenging benchmarks. More remarkably, IterResearch narrows the performance gap with frontier proprietary systems, even surpassing some on these benchmarks. Furthermore, our work reveals three fundamental insights about deep-research agents. First, our iterative paradigm unlocks extreme **interaction scaling**—a capability theoretically extensible to infinite depths yet structurally infeasible for current mono-contextual approaches. To our knowledge, we are the first to successfully extend agents to 2048 interactions with only 40K context length, exhibiting dramatic performance improvements ($3.5\% \rightarrow 42.5\%$) as maximum interactions increase from 2 to 2048, suggesting that the perceived difficulty of long-horizon tasks may stem from insufficient exploration capacity. Second, we observe **cross-paradigm knowledge transfer**: trajectories generated by IterResearch significantly enhance mono-contextual agents, demonstrating that our paradigm induces superior exploration behaviors that create high-quality training signals transferable even across paradigmatically different approaches. Third, our iterative paradigm serves as an **effective prompting strategy**: without any training, simply applying it to frontier models yields substantial improvements over the standard mono-contextual approach, ReAct (Yao et al., 2023), particularly on long-horizon tasks (+12.7-19.2pp on BrowseComp), revealing that IterResearch offers a *model-agnostic* solution to long-horizon reasoning. These results confirm the effectiveness of our iterative paradigm in enabling both deeper exploration and higher-quality reasoning in long-horizon scenarios.

In summary, our main contributions can be summarized as follows:

- We propose IterResearch, a novel iterative deep-research paradigm that revisits long-horizon research through an MDP-based formalism, maintaining sustained reasoning capacity through periodic synthesis and an evolving report memory—eliminating the context suffocation and noise contamination that plague mono-contextual approaches.

- We employ Efficiency-Aware Policy Optimization (EAPO) which adapts geometric discounted rewards that incentivize efficient exploration and utilizes adaptive downsampling for stable distributed training, enabling effective learning from our paradigm's unique trajectory structure.

- We demonstrate IterResearch's exceptional capabilities and broader impact: (1) achieving an average 14.5 pp improvement across six challenging benchmarks; (2) exhibiting interaction scaling to 2048 interactions with dramatic performance gains; (3) enabling cross-paradigm knowledge transfer to enhance mono-contextual agents; (4) providing a model-agnostic prompting strategy that significantly improves frontier models on long-horizon tasks without training.

## 2 RELATED WORK

**Retrieval-Augmented Generation (RAG).** RAG is a crucial approach to overcome knowledge limitations of large language models (LLMs) by integrating external information sources (Nakano et al., 2021; Yu et al., 2024; Asai et al., 2024; Wei et al., 2025b; Chen et al., 2025a; Jin et al., 2025; Song et al., 2025; Zheng et al., 2025b). However, traditional RAG methods are typically confined to static retrieval environments, such as Wikipedia, with limited exploration spaces, making them inadequate for complex, long-horizon reasoning tasks that require dynamic information gathering.

**Deep Research.** Recent advances in deep research (OpenAI, 2025a; Google, 2025; xAI, 2025; MoonshotAI, 2025) have transcended RAG's limitations by deploying autonomous agents in real-world environments, demonstrating remarkable capabilities in navigating complex web environments and synthesizing information from diverse sources. However, existing open-source methods (Li et al., 2025b;c; Tao et al., 2025; Li et al., 2025a) predominantly adopt a *mono-contextual paradigm*, continuously appending all retrieved information and reasoning steps to a single expanding context. This linear accumulation leads to progressive workspace suffocation and irreversible noise contamination, limiting their effectiveness in long-horizon tasks. In contrast, our IterResearch reimagines deep research by adopting an MDP-based architecture with a iterative workspace reconstruction mechanism, eliminating accumulation-induced degradation and enabling sustained reasoning capacity at arbitrary research depths—a critical advantage absent in existing approaches.

## 3 METHODOLOGY

In this section, we detail IterResearch, which extends the Markov Decision Process framework to deep research through iterative workspace reconstruction (§3.1), as illustrated in Figure 2. Then, we further introduce Efficiency-Aware Policy Optimization for training (§3.2).

### 3.1 ITERATIVE DEEP-RESEARCH PARADIGM

#### 3.1.1 MDP-INSPIRED FORMULATION

To address the limitations of mono-contextual paradigms, we adopt an MDP-inspired formalism to structure long-horizon research, defined by the tuple $\langle \mathcal{S}, \mathcal{D}, \mathcal{E}, \mathcal{T}, R \rangle$. This formulation explicitly separates internal thought updates from external environmental interactions to enforce state independence.

- **State Space** $\mathcal{S}$: Each state $s_t = (q, \mathcal{M}_t, \{a_{t-1}, \text{TR}_{t-1}\})$ represents the agent's explicit workspace. $s_t$ serves as a synthesized representation of the research history, comprising: (1) the constant question $q$; (2) the evolving report $\mathcal{M}_t$, which acts as a compressed memory of prior validated findings; (3) the immediate context $\{a_{t-1}, \text{TR}_{t-1}\}$ from the previous step.

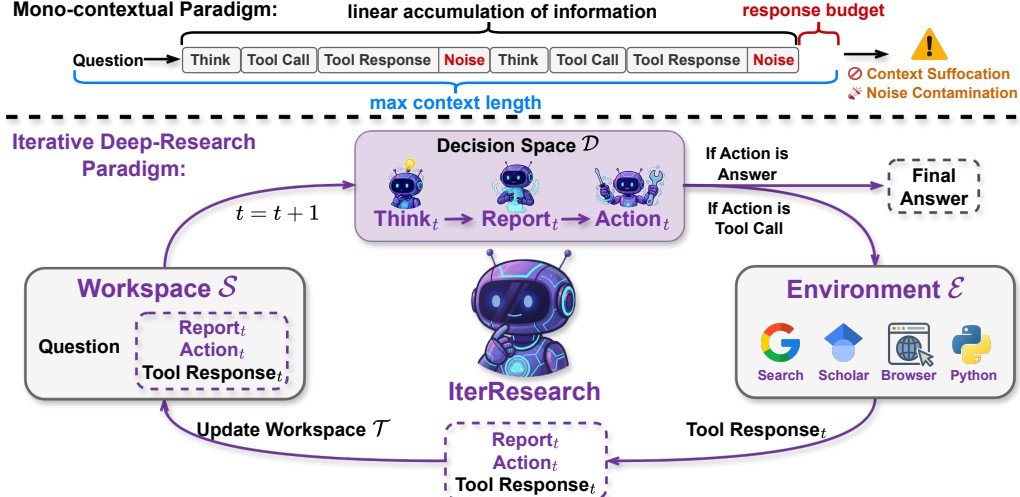

Figure 2: **(Top)** The mono-contextual approach linearly accumulates all information into a single, ever-expanding context, leading to context suffocation and noise contamination. **(Bottom)** IterResearch models deep research with iterative workspace reconstruction. Each round begins with a reconstructed workspace $s_t$ containing the question, an evolving report $\mathcal{M}_t$, and immediate context. The agent generates structured decisions $d_t = (\textit{Think}, \textit{Report}, \textit{Action})$ and interacts with environment $\mathcal{E}$. The transition function $\mathcal{T}$ reconstructs the workspace, maintaining an iterative state structure while preventing context bloat and enabling sustained reasoning and information-seeking.

- **Decision Space** $\mathcal{D}$: At each state $s_t$, the policy $\pi$ generates a composite decision $d_t$ consisting of distinct internal thought and external actions:

$$d_t = [\underbrace{\text{Think}_t, \mathcal{M}_{t+1}}_{\text{Internal Thought}}, \underbrace{a_t}_{\text{External Action}}] \sim \pi(\cdot|s_t) \qquad (1)$$

  Here, $\mathcal{M}_{t+1}$ represents an active memory update generated by the agent, while $a_t$ is the actual interaction with the environment or the final answer.
- **Environment** $\mathcal{E}$ and **Reward** $R$: The environment executes action $a_t$ and returns a stochastic response $\text{TR}_t \sim \mathcal{E}(\cdot|a_t)$. The reward function $R(s_t, a_t)$ equals 1 if and only if the action $a_t$ is a terminal answer evaluated as correct by the oracle, and 0 otherwise.
- **Transition Function** $\mathcal{T}$: The state transition is deterministic given the decision and environmental response. It explicitly reconstructs the workspace rather than appending to history:

$$s_{t+1} = \mathcal{T}(s_t, d_t, \text{TR}_t) = (q, \mathcal{M}_{t+1}, \{a_t, \text{TR}_t\}) \qquad (2)$$

  This reconstruction mechanism ensures that $s_{t+1}$ depends solely on the current state $s_t$ (via $d_t$) and the immediate environment feedback $\text{TR}_t$, strictly adhering to the MDP intuition.

The complete research process of IterResearch can be formalized as a sequence of state transitions driven by the agent policy $\pi$:

$$\begin{cases} \text{Policy Step:} & (\text{Think}_t, \mathcal{M}_{t+1}, a_t) \sim \pi(\cdot|s_t) \\ \text{Environment Step:} & \text{TR}_t \sim \mathcal{E}(\cdot|a_t) \\ \text{State Update:} & s_{t+1} \leftarrow (q, \mathcal{M}_{t+1}, \{a_t, \text{TR}_t\}) \end{cases} \qquad (3)$$

where $\text{TR}_t = \mathcal{E}(a_t)$, initial state $s_0 = (q, \mathcal{M}_0, \emptyset)$ with empty report $\mathcal{M}_0$. The iterative process generates a trajectory $\tau = \{(s_0, d_0, \text{TR}_0), (s_1, d_1, \text{TR}_1), \ldots, (s_T, d_T)\}$ terminating when $a_T = $ `answer`. Unlike mono-contextual approaches where the state dimension grows linearly ($|s_t| \propto t$), our formulation ensures $|s_t| \approx O(1)$, enabling theoretically unbounded exploration steps.

### 3.1.2 ITERATIVE WORKSPACE RECONSTRUCTION

The cornerstone of our paradigm is workspace reconstruction, which fundamentally departs from traditional linear accumulation approaches (Li et al., 2025c;a; Tao et al., 2025). While existing

methods suffer from $O(t)$ context growth leading to inevitable performance degradation, we introduce a principled reconstruction mechanism that maintains bounded workspace complexity while preserving complete task-relevant information through selective compression.

At round $t$, the workspace $s_t$ contains only three essential components: (1) the question $q$, providing the constant objective; (2) the evolving report $\mathcal{M}_t$, serving as compressed memory of all critical findings; and (3) the immediate context $\{a_{t-1}, \text{TR}_{t-1}\}$ from the last interaction. The key insight is that the report $\mathcal{M}_{t+1}$ is *naturally generated* by the LLM as part of its structured decision output $d_t = (\text{Think}_t, \mathcal{M}_{t+1}, a_t)$. This natural flow leverages the LLM's inherent capabilities for information compression and relevance filtering, without requiring explicit algorithmic intervention.

As shown in Eq. 3, the transition function $\mathcal{T}$ implements strategic forgetting by reconstructing the workspace at each round. The historical trajectory $(s_0, d_0, \text{TR}_0, ..., s_{t-1}, d_{t-1}, \text{TR}_{t-1})$ is deliberately discarded, with only the synthesized knowledge preserved in $\mathcal{M}_{t+1}$. This design ensures a constant workspace regardless of trajectory length, in stark contrast to mono-contextual approaches:

$$\underbrace{s_t^{\text{mono}} = [q, a_0, \text{TR}_0, ..., a_{t-1}, \text{TR}_{t-1}]}_{\text{Mono-contextual: } \mathcal{O}(t) \text{ growth}} \quad \text{vs.} \quad \underbrace{s_t^{\text{iter}} = (q, \mathcal{M}_t, \{a_{t-1}, \text{TR}_{t-1}\})}_{\text{IterResearch (Ours): } \mathcal{O}(1) \text{ constant}} \tag{4}$$

Through the iterative workspace, the agent maintains consistent reasoning capacity throughout the research process, avoiding the performance degradation that inevitably occurs when context windows approach their limits. Furthermore, through end-to-end training (§3.2), the agent progressively learns to synthesize reports that effectively filter noise and preserve essential information. Thus, irrelevant information or errors from early rounds cannot directly propagate to future decisions—they must first pass through the agent's synthesis to be incorporated into the report. This selective retention aligns with the MDP formalism: the current state $s_{t+1}$ contains all decision-relevant information, making the full history unnecessary for optimal decision-making. The transformative impact of this design manifests in ***interaction scaling***. While mono-contextual approaches typically fail or degrade severely beyond dozens of interactions due to context limitations, our IterResearch enables theoretically unbounded exploration, sustaining consistent reasoning quality at arbitrary depths. This scaling capability, empirically validated through experiments with up to 2048 interactions (§ 4.4), fundamentally expands the scope of problems that deep-research agents can tackle.

## 3.2 EFFICIENCY-AWARE POLICY OPTIMIZATION

### 3.2.1 DISCOUNTED REWARD SHAPING FOR EFFICIENCY

While the iterative workspace reconstruction ensures scalable exploration, a critical question remains: how can we train agents to not just explore deeply, but to do so *efficiently*? We now address this challenge by employing an efficiency-aware training strategy.

In deep research tasks, the agent receives a binary reward signal $R_T \in \{0, 1\}$ only upon termination, where $R_T = 1$ if the final answer is correct and 0 otherwise. This terminal-only reward stems from the inherent difficulty of evaluating intermediate research steps—it is challenging to determine the value of any particular search query or exploratory action (Chen et al., 2025a).

However, this sparse signal alone is insufficient for guiding efficient learning, ***as it treats all successful trajectories equally regardless of their computational cost***. An agent that arrives at the correct answer in 5 well-chosen steps should be preferred over one that requires 20 steps of meandering exploration, even if both ultimately succeed. This efficiency consideration is not merely about computational resources: in real-world deployment, each interaction incurs API costs, and unnecessary exploration can lead to increased latency. To address these issues, we adapt the standard episodic reward formulation from Sutton & Barto (2018). By utilizing geometric discounting, we create an implicit efficiency pressure:

$$r_t = \gamma^{T-t} \cdot R_T, \quad \gamma \in (0, 1) \tag{5}$$

where $T$ is the terminal step, $t$ is the current step, and $\gamma$ is the discount factor. This exponential decay creates an implicit efficiency pressure: actions contributing to earlier task completion receive proportionally higher rewards, naturally incentivizing more direct exploration strategies while maintaining the simplicity of terminal-only evaluation.

### 3.2.2 Policy Optimization with Multi-Round Trajectories

A distinctive feature of our iterative paradigm is that each trajectory naturally decomposes into multiple independent training samples (one per round), whereas one trajectory typically yields a single training sample in mono-contextual approaches. Specifically, for each question $q$, we perform $G$ rollouts generating $G$ independent trajectories. Each trajectory $\tau_i$ unfolds over $T_i$ rounds, where round $t$ produces a state-decision pair $(s_{i,t}, d_{i,t})$ following our MDP formulation (Eq. 3).

This yields a rich training corpus $\mathcal{C} = \{(s_{i,t}, d_{i,t}, r_{i,t}) : i \in [1, G], t \in [1, T_i]\}$ with $\sum_{i=1}^{G} T_i$ samples, far exceeding the $G$ trajectory-level samples from traditional approaches. While this paradigm significantly enriches training data, the variable sample count across questions requires careful handling for distributed training. We address this through *adaptive downsampling* that reduces the training corpus to the largest multiple of data parallel (DP) size:

$$|\mathcal{C}_{\text{train}}| = \left\lfloor \frac{|\mathcal{C}|}{\text{DP}_{\text{size}}} \right\rfloor \times \text{DP}_{\text{size}} \tag{6}$$

This approach ensures minimal data loss (typically $< 1\%$ of samples) while maintaining uniform sampling across trajectories. To optimize IterResearch, we combine these efficiency-aware rewards and adaptive downsampling into a unified training strategy (EAPO), implemented on top of the Group Sequence Policy Optimization (GSPO) algorithm (Zheng et al., 2025a), enabling stable training on variable-length trajectories:

$$\mathcal{J}(\theta) = \mathbb{E}_{q \sim \mathcal{Q}, \mathcal{C}_{\text{train}} \sim \pi_{\theta_{\text{old}}}(\cdot|q)} \left[ \frac{1}{|\mathcal{C}_{\text{train}}|} \sum_{i=1}^{G} \sum_{t=1}^{T_i} \min(\rho_{i,t}(\theta)\hat{A}_{i,t}, \text{clip}(\rho_{i,t}(\theta), 1 - \varepsilon, 1 + \varepsilon)\hat{A}_{i,t}) \right] \tag{7}$$

where all $\sum_{i=1}^{G} T_i$ rounds from the $G$ trajectories for question $q$ form *one group*, with normalized advantages computed across all samples within this group $\hat{A}_{i,t} = \frac{r_{i,t} - \mu_r}{\sigma_r}$, $\mathcal{Q}$ is the training set, and $\rho_{i,t}(\theta)$ is the importance ratio based on sequence likelihood (Zheng et al., 2023).

## 4 Experiments

### 4.1 Experimental Setup

**Datasets.** To rigorously assess the effectiveness of our IterResearch, we evaluate on six challenging benchmarks including **Humanity's Last Exam (HLE)** (Phan et al., 2025), **BrowseComp** (Wei et al., 2025a), **BrowseComp-zh** (Zhou et al., 2025a), **GAIA** (Mialon et al., 2023), **Xbench-DeepSearch** (Xbench-Team, 2025), **SEAL-0** (Pham et al., 2025). These benchmarks comprehensively assess the essential capabilities for effective deep research in multi-step tool use, web navigation, complex reasoning, long-horizon information-seeking, and cross-lingual synthesis.

**Baselines.** We comprehensively compare our IterResearch against state-of-the-art methods including: (1) **Direct Inference**: We evaluate frontier LLMs including GPT-4o and GPT-4.1 (Hurst et al., 2024), o4-mini (OpenAI, 2025b), and DeepSeek-R1-0528 (Guo et al., 2025). (2) **Proprietary Deep-Research System**: We compare with commercial deep-research systems including OpenAI's Deep Research (OpenAI, 2025a), Perplexity Research (Perplexity, 2025), Gemini Deep Research (Google, 2025), Grok3-ResearchSearch (xAI, 2025), and Kimi-Researcher (MoonshotAI, 2025). (3) **Open-source Agents**: Recent open-source deep-research agents including Search-o1 (Li et al., 2025b), WebThinker (Li et al., 2025c), WebDancer (Wu et al., 2025), WebSailor (Li et al., 2025a), Asearcher (Gao et al., 2025), and MiroThinker (MiroMindAI, 2025).

**Implementation Details.** We implement our IterResearch using Qwen3-30B-A3B (Yang et al., 2025) as the backbone model, considering both model performance and computational efficiency. Our training follows a two-stage process: we first employ rejection sampling fine-tuning (RFT) (Yuan et al., 2023) to equip the model with our iterative deep-research paradigm capabilities, then apply reinforcement learning to further enhance its search strategy and reasoning abilities. For brevity, we provide comprehensive training details and hyperparameters in Appendix C.3.

Table 1: Main results across six deep-research benchmarks. We report accuracy (%) for all metrics. The best results are in **bold**, and the second best among open-source agents are underlined.

| Model | Tools | HLE | BC | BC-zh | GAIA | Xbench-DS | SEAL-0 |
|---|---|---|---|---|---|---|---|
| *Direct Inference* | | | | | | | |
| GPT-4o | ✗ | 2.3 | 0.6 | 6.2 | 17.5 | - | - |
| GPT-4.1 | ✗ | 4.9 | 1.5 | 14.4 | 22.3 | - | - |
| o4-mini | ✗ | **18.9** | **6.1** | 15.2 | **33.3** | 60.0 | 4.5 |
| DeepSeek-R1-0528 | ✗ | 17.7 | 2.0 | **26.3** | 16.5 | - | **5.4** |
| *Proprietary Deep-Research System* | | | | | | | |
| OpenAI DeepResearch | ✓ | 26.6 | **51.5** | **42.9** | **67.4** | - | - |
| Perplexity Research | ✓ | 21.1 | - | 22.6 | - | - | - |
| Gemini DeepResearch | ✓ | **26.9** | - | - | - | 50.0 | - |
| Grok3-ResearchSearch | ✓ | - | - | 12.9 | - | 50.0 | - |
| Kimi-Researcher | ✓ | **26.9** | - | - | - | **69.0** | **36.0** |
| *Open-source Agents* | | | | | | | |
| Search-o1-QwQ | ✓ | 5.4 | 2.8 | 17.9 | 39.8 | 40.3 | - |
| WebThinker-QwQ | ✓ | 6.8 | 2.8 | 7.3 | 48.5 | 32.8 | - |
| WebDancer-QwQ | ✓ | 7.6 | 3.8 | 18.0 | 51.5 | 40.0 | 20.7 |
| Asearcher-Web-QwQ | ✓ | 12.5 | 5.2 | 15.6 | 52.8 | 42.1 | - |
| WebSailor-32B | ✓ | 9.6 | 10.5 | 25.5 | 53.2 | 53.3 | 16.2 |
| WebSailor-72B | ✓ | 9.8 | 12.0 | 30.1 | 55.4 | 55.0 | 19.8 |
| MiroThinker-14B$_{v0.2}$ | ✓ | 20.0 | 14.1 | 26.6 | 62.1 | 47.0 | - |
| MiroThinker-32B$_{v0.2}$ | ✓ | 19.1 | 17.2 | 29.4 | 64.1 | 56.0 | - |
| **IterResearch-30B-A3B** | ✓ | **28.8** $\pm$ 0.5 | **37.3** $\pm$ 0.7 | **45.2** $\pm$ 1.6 | **72.8** $\pm$ 2.3 | **71.0** $\pm$ 0.5 | **39.6** $\pm$ 0.8 |
| + Improvement | | ↑ 8.8 | ↑ 20.1 | ↑ 15.8 | ↑ 8.7 | ↑ 15.0 | ↑ 18.9 |

## 4.2 MAIN RESULTS

Table 1 presents the comprehensive evaluation results across six challenging benchmarks. **First,** IterResearch outperforms all existing open-source agents, with an average margin of 14.5 percentage points across the six benchmarks. More remarkably, it demonstrates competitive or superior performance compared to proprietary deep-research systems—surpassing OpenAI's DeepResearch on HLE and BrowseComp-zh, while achieving comparable results on BrowseComp and GAIA. These results confirm that our iterative paradigm successfully bridges the gap between open-source and commercial systems. **Second**, the consistent improvements across benchmarks with distinct characteristics validate our core design principles. On *information-seeking benchmarks* requiring extensive web navigation (BrowseComp, BrowseComp-zh, SEAL-0), our method demonstrates substantial advantages over mono-contextual baselines. These tasks particularly suffer from context suffocation in traditional approaches, as agents must navigate through numerous web pages while synthesizing vast amounts of information. Our workspace reconstruction mechanism maintains consistent reasoning capacity by strategically compressing findings into the evolving report, preventing the inevitable degradation that plagues mono-contextual methods. On *complex reasoning benchmarks* demanding deep analytical capabilities (HLE, GAIA, Xbench-DS), the advantage stems from our ability to mitigate noise contamination. While mono-contextual approaches irreversibly accumulate errors and irrelevant information throughout their trajectories, our iterative paradigm provides natural breakpoints for filtering noise through periodic synthesis. The evolving report preserves only validated findings while discarding exploratory dead-ends, enabling more focused reasoning in subsequent rounds. These consistent improvements across diverse task types demonstrate that the iterative deep-research paradigm provides a principled solution to the fundamental limitations of linear information accumulation

## 4.3 ABLATION STUDY

To thoroughly understand the contributions of our approach, we conduct comprehensive ablation studies examining both the effectiveness of our Efficiency-Aware Policy Optimization (EAPO) and the fundamental advantages of our iterative paradigm over traditional mono-contextual approaches.

Table 2: Ablation studies on training methodology and paradigm design. The paradigm ablation uses identical training data and external environment to ensure fair comparison.

| | HLE | BC | BC-zh | GAIA | Xbench-DS | SEAL-0 | Avg |
|---|---|---|---|---|---|---|---|
| Ablation on Methodology | | | | | | | |
| IterResearch-EAPO | 28.8 | 37.3 | 45.2 | 72.8 | 71.0 | 39.6 | 49.1 |
| IterResearch-GSPO | 28.2 | 38.3 | 45.6 | 70.9 | 67.0 | 39.6 | 48.3 |
| IterResearch-SFT | 25.3 | 34.9 | 40.8 | 68.9 | 65.0 | 37.8 | 45.5 |
| Ablation on Paradigm (Cross-Paradigm Knowledge Transfer) | | | | | | | |
| Mono-Agent | 18.7 | 25.4 | 34.6 | 62.1 | 55.0 | 23.4 | 36.5 |
| Mono-Agent + Iter | 25.4 | 30.1 | 40.4 | 63.1 | 62.0 | 30.6 | 41.9 |
| + Improvement | ↑ 6.7 | ↑ 4.7 | ↑ 5.8 | ↑ 1.0 | ↑ 7.0 | ↑ 7.2 | ↑ 5.4 |

(1) *Effectiveness of Efficiency-Aware Policy Optimization.* The upper section of Table 2 demonstrates the impact of our EAPO compared to standard GSPO and SFT. Analysis of average interactions reveals that EAPO requires 18.04 turns, compared to GSPO's 19.13 turns and SFT's 16.45 turns. While EAPO and GSPO achieve comparable accuracy across benchmarks, the critical distinction emerges in interaction efficiency: **EAPO reduces average interactions by 5.7% while maintaining or improving accuracy**. This validates our core hypothesis that geometric discounted rewards successfully incentivize the discovery of more efficient research strategies—agents learn to reach correct conclusions through more focused, deliberate exploration rather than exhaustive searching.

(2) *Superiority of the Iterative Paradigm.* To rigorously validate our paradigm's advantages, we conduct a controlled comparison using *identical training data* across different paradigms. The middle section of Table 2 reveals striking performance gaps: our iterative paradigm outperforms the mono-contextual baseline (**Mono-Agent**) by an average of 12.6 percentage points across all benchmarks, with particularly dramatic improvements on long-horizon information-seeking tasks (BC: +11.8%, BC-zh: +10.6%). Notably, to ensure the mono-contextual agent operates at its optimal capacity and mitigate the inevitable context accumulation issues inherent to its design, we deliberately equipped it with a substantially larger context window (64K vs. our 40K tokens). This substantial performance gap persists despite providing the mono-contextual approach with more context length, which confirms our theoretical analysis: **workspace suffocation fundamentally limits mono-contextual approaches—simply expanding the context window cannot resolve this limitation**. In contrast, our workspace reconstruction mechanism maintains consistent reasoning quality at arbitrary depths through strategic information compression and filtering, enabling effective handling of long-horizon tasks that overwhelm traditional approaches regardless of their context size.

(3) *Cross-Paradigm Knowledge Transfer.* An unexpected yet significant finding emerges: **trajectories generated by our iterative paradigm can enhance mono-contextual agents when incorporated into their training data**. As shown in the bottom rows of Table 2, augmenting Mono-Agent with iterative-paradigm data while maintaining total data volume (**Mono-Agent + Iter**) yields consistent improvements across most benchmarks, with an average gain of 5.4 percentage points. The fact that trajectories generated through our iterative paradigm can enhance mono-contextual agents indicates that our paradigm induces superior research behaviors that create higher-quality training signals, partially transferable even across paradigmatically different approaches.

## 4.4 SCALING ON INTERACTION

A fundamental advantage of our iterative paradigm is its ability to maintain consistent performance at arbitrary interaction depths—a property critical for tackling genuinely complex long-horizon tasks that may require extensive exploration. To empirically validate this capability, we conduct scaling experiments on BrowseComp (200 subset), the most interaction-intensive benchmark in our evaluation suites. Figure 3 presents our scaling analysis as we exponentially increase the maximum allowed turns from 2 to 2048, a range that would be computationally prohibitive for mono-contextual approaches due to context window limitations. Two key insights emerge from these results:

**First, performance scales gracefully with interaction budget.** Accuracy improves from 5.5% with only 2 turns to 50.1% at 2048 turns, with the steepest gains occurring between $2^4$ and $2^7$ turns. This demonstrates that complex information-seeking tasks genuinely benefit from extended exploration— a capability that mono-contextual approaches cannot provide due to inevitable context overflow. Notably, 2048 turns represents an extreme challenge that is currently infeasible for mono-contextual agents due to catastrophic context accumulation,

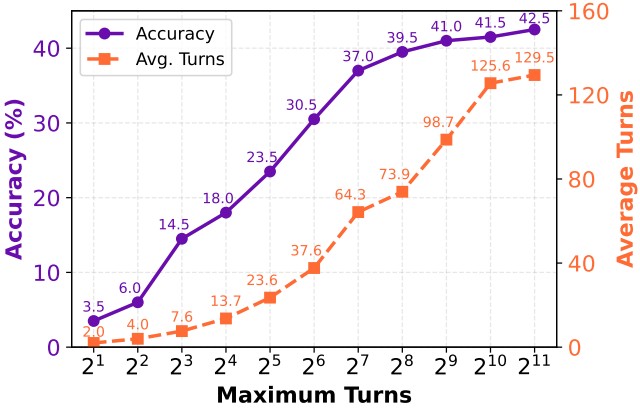

Figure 3: Interaction Scaling.

yet our approach operates smoothly within its constant 40K token workspace through iterative state reconstruction. **Second, the agent learns intelligent resource allocation.** Despite having access to 2048 turns, the agent uses only 80.1 turns on average, indicating adaptive termination once sufficient information is gathered rather than exhaustively consuming the budget. Notably, the growth pattern of average turns mirrors the accuracy curve—both increase rapidly in the $2^4$-$2^7$ range before plateauing—suggesting that exploration depth naturally aligns with task complexity. This sublinear growth in average turns (compared to exponentially increasing budget) demonstrates that the agent develops increasingly efficient search strategies as more interactions become available, rather than simply extending existing patterns.

## 4.5 ITERRESEARCH AS A EFFECTIVE PROMPTING STRATEGY IN LONG-HORIZON TASKS

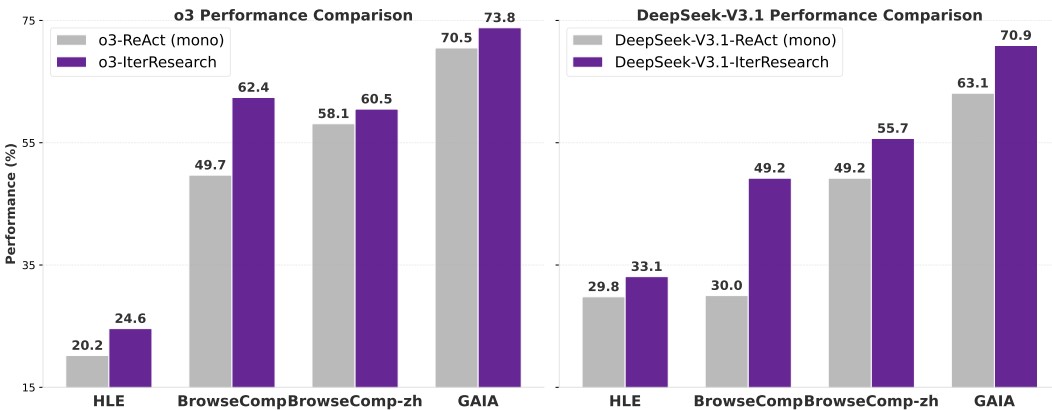

Figure 4: Performance comparison between IterResearch and ReAct as Prompting Strategies.

Having demonstrated IterResearch's effectiveness as a trained agent, we investigate whether our iterative paradigm can serve as an effective *prompting strategy for long-horizon tasks* without any training. We compare with ReAct (Yao et al., 2023), the prevailing mono-contextual prompting paradigm, using frontier models o3 (OpenAI, 2025b) and DeepSeek-V3.1 (DeepSeek, 2025).

Figure 4 reveals that **IterResearch consistently outperforms ReAct across all benchmarks**, with particularly dramatic improvements on the most challenging long-horizon task BrowseComp (o3: +12.7pp, DeepSeek: +19.2pp). These gains validate two key insights: (1) The iterative paradigm with workspace reconstruction provides a more effective cognitive structure for long-horizon reasoning, enabling models to maintain focus through periodic synthesis rather than drowning in accumulated context. (2) The paradigm's benefits are *model-agnostic*—both o3 and DeepSeek model architectures exhibit substantial improvements, suggesting that our approach addresses fundamental limitations in how current models handle extended reasoning chains rather than model-specific

weaknesses. The improvements peak on BrowseComp—the most exploration-intensive benchmark—confirming that our paradigm's advantages scale with task horizon length, making it particularly valuable for complex real-world problems.

## 5 CONCLUSION

In this work, we presented IterResearch, a novel iterative deep-research paradigm that addresses the context suffocation and noise contamination plaguing mono-contextual approaches in long-horizon tasks. By iterative workspace reconstruction and developing Efficiency-Aware Policy Optimization for effective training, we achieved substantial improvements over existing agents (average +14.5pp across six benchmarks). Furthermore, our experiments reveal three transformative insights: this iterative paradigm enables unprecedented interaction scaling to 2048 interactions with dramatic performance gains (3.5% to 42.5%), serves as an effective prompting strategy that improves frontier models by up to 19.2pp, and induces superior exploration behaviors transferable across different paradigms. These findings establish that iteration with strategic synthesis, rather than accumulation, is fundamental to conquering long-horizon reasoning challenges, providing both a powerful agent architecture and a versatile framework applicable across different models and paradigms.

### ACKNOWLEDGMENTS

This work was supported by Alibaba Research Intern Program. This paper was partially supported by the National Natural Science Foundation of China No. 92470205 and Beijing Major Science and Technology Project under Contract No. Z251100008425002.

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

## A  ADDTIONAL RELATED WORK

**Memory Mechanisms in LLMs.** Memory mechanisms have emerged as a critical component for extending LLM capabilities beyond single-turn interactions (Du et al., 2025; Zhang et al., 2025; Chen et al., 2025b). While early works explored explicit memory architectures with separate storage and retrieval modules (Yang et al., 2024), recent approaches have focused on memory management for LLM agents. MemoryLLM (Hu et al., 2025) and MEM1 (Zhou et al., 2025b) investigate how agents can learn to synthesize and utilize memory across multi-turn interactions, while Memory-R1 (Yan et al., 2025) employs reinforcement learning to train agents for adaptive memory management. MemAgent (Yu et al., 2025) and MemOS (Li et al., 2025d) further advance this direction by introducing memory operating systems that unify representation, scheduling, and evolution of memories as manageable system resources. However, these memory-centric approaches primarily focus on explicit memory module design or retrieval optimization within fixed context windows, fundamentally differing from our approach. IterResearch naturally integrates memory through the evolving report $\mathcal{M}_t$ within our Markovian workspace reconstruction—rather than maintaining separate memory modules or databases, our report serves as a compressed, task-focused memory that is seamlessly updated through the agent's structured decisions. This design eliminates the overhead of explicit memory management while ensuring that memory evolution is intrinsically aligned with the research trajectory, enabling more efficient and coherent long-horizon exploration.

**Iterative Reasoning and Convergent Paradigms.** Concurrently, similar iterative paradigms have emerged in different domains, most notably in algorithmic discovery with systems like AlphaEvolve (Novikov et al., 2025). AlphaEvolve, designed for coding tasks, shares a strong conceptual architecture with IterResearch: it maintains a main objective (the task), an "evolving report" of high-level ideas (analogous to our $\mathcal{M}_t$), and an immediate context (the code). Despite this architectural parallel, the domains are fundamentally different. AlphaEvolve operates on structured code and receives verification from unit tests, whereas IterResearch is designed for the unstructured, noisy, and dynamic environment of web research. We believe this connection strengthens our core thesis. The "convergent evolution" of two distinct, long-horizon domains (algorithmic discovery and deep web research) independently arriving at an iterative synthesis paradigm with an evolving report strongly suggests this is a fundamental and highly effective solution to the limitations of mono-contextual reasoning. It indicates that our IterResearch paradigm is not a niche method, but a general and advanced architecture for complex, long-horizon AI.

## B  MORE ANALYSIS

### B.1  THEORETICAL MOTIVATION: EFFICIENCY THROUGH DISCOUNTING

The discounted reward formulation in Eq. 5 elegantly encodes a preference for efficiency that emerges naturally from the MDP framework. To illustrate this, consider two successful research trajectories for the same question: trajectory $\tau_A$ reaching the correct answer in $T_A = 5$ steps, and trajectory $\tau_B$ requiring $T_B = 20$ steps.

Under our discounting scheme with $\gamma = 0.995$, each step in the trajectories receives different rewards based on its temporal distance from the terminal state. For any intermediate step $t$, the rewards are:

$$r_t^A = \gamma^{T_A - t} \cdot R_T = \gamma^{5-t} \tag{8}$$

$$r_t^B = \gamma^{T_B - t} \cdot R_T = \gamma^{20-t} \tag{9}$$

This creates a fundamental learning signal: **earlier steps in shorter trajectories receive substantially higher rewards than corresponding steps in longer trajectories**. To illustrate the magnitude of this difference, consider the reward at step $t = 3$:

$$r_3^A = \gamma^{5-3} = \gamma^2 \approx 0.99 \tag{10}$$

$$r_3^B = \gamma^{20-3} = \gamma^{17} \approx 0.918 \tag{11}$$

The 7.8% reward difference for the same step position creates a strong gradient that guides the policy toward more efficient research strategies. This consistent multiplicative advantage across all shared steps systematically guides the policy toward discovering more efficient research strategies.

- **Redundant exploration**: Searching for similar information multiple times delays progress, with each redundant step reducing future rewards by factor $\gamma$
- **Circular reasoning**: Revisiting previously explored hypotheses without new insights wastes steps, exponentially diminishing the trajectory's total return
- **Unfocused browsing**: Following tangential information that doesn't contribute to the final answer accumulates geometric penalties

Importantly, this efficiency incentive emerges *without any explicit length penalty or auxiliary objectives*—it is an inherent property of geometric discounting applied to our MDP formulation. The discount factor $\gamma$ serves as a single hyperparameter that controls the trade-off between exploration thoroughness and efficiency: values closer to 1 allow more exploratory behavior, while smaller values create stronger pressure for direct problem-solving. Our empirical choice of $\gamma = 0.995$ strikes a balance that permits necessary exploration while maintaining sufficient efficiency pressure, as validated by the 5.7% reduction in average trajectory length observed in our ablation studies (Table 2).

### B.2 COMPUTATIONAL COMPLEXITY ANALYSIS.

Unlike mono-contextual approaches where context size grows as $O(t \cdot |\text{TR}|)$ with $t$ rounds and average response size $|\text{TR}|$, our algorithm maintains a constant workspace size of $O(|\mathcal{M}| + |\text{TR}|)$, where $|\mathcal{M}|$ is the report size bounded by design through the agent's learned synthesis behavior. **This ensures consistent computational efficiency regardless of the research depth.** Table 3 provides a detailed complexity comparison. In the Table, $t$ is the number of rounds, $|\text{TR}|$ is the average tool response size, $|\mathcal{M}|$ is the bounded report size, and $L$ is the model's context limit. The key distinctions are:

Table 3: Computational complexity comparison between paradigms.

| Metric | Mono-contextual | IterResearch (Ours) |
|---|---|---|
| Used Context Size | $O(t \cdot |\text{TR}|)$ | $O(|\mathcal{M}| + |\text{TR}|)$ |
| Attention Computation | $O((t \cdot |\text{TR}|)^2)$ | $O((|\mathcal{M}| + |\text{TR}|)^2)$ |
| Effective Reasoning Window | $O(\max(0, L - t \cdot |\text{TR}|))$ | $O(L - |\mathcal{M}| - |\text{TR}|)$ |
| Maximum Rounds | $O(L/|\text{TR}|)$ | $O(\infty)$ (theoretically unbounded) |

- **Used Context Size**: Mono-contextual approaches accumulate all past responses, growing linearly with rounds until reaching the context limit. Our approach maintains constant size through workspace reconstruction, with the report $\mathcal{M}$ serving as a compressed memory that synthesizes all essential findings.

- **Attention Computation**: The quadratic attention cost becomes prohibitive for mono-contextual approaches as $t$ increases, with complexity scaling as $O((t \cdot |\text{TR}|)^2)$. Our bounded workspace ensures consistent computational cost of $O((|\mathcal{M}| + |\text{TR}|)^2)$ per round, independent of trajectory length.

- **Effective Reasoning Window**: In mono-contextual approaches, the available context for new reasoning diminishes as $\max(0, L - t \cdot |\text{TR}|)$, eventually reaching zero when accumulated history exhausts the context limit. Our approach maintains a consistent reasoning window of $L - |\mathcal{M}| - |\text{TR}|$ across all rounds, ensuring sustainable reasoning capacity throughout the research process.

- **Maximum Rounds**: Mono-contextual approaches face a hard limit of approximately $L/|\text{TR}|$ rounds before context overflow. In contrast, our iterative paradigm is theoretically unbounded—as long as $|\mathcal{M}| + |\text{TR}| < L$ (which is maintained through report synthesis), the agent can continue exploration indefinitely.

These complexity advantages become critical in long-horizon tasks: while mono-contextual approaches face inevitable failure when $t \cdot |\text{TR}| > L$ (context overflow), our approach can theoretically extend to arbitrary depths. This theoretical advantage translates to practical benefits, as empirically demonstrated in our scaling experiments (Figure 3), where we successfully extend agents to 2048 interactions using only 40K context length—a feat structurally impossible for mono-contextual approaches.

The constant complexity also ensures predictable resource consumption: each round requires approximately the same computational resources regardless of position in the trajectory, enabling better resource planning and allocation in deployment scenarios. This predictability, combined with the unbounded exploration capability, makes our iterative paradigm particularly suitable for genuinely complex research tasks that may require extensive investigation.

### B.3 EXTRAPOLATION BEYOND TRAINING HORIZON

A remarkable property of our iterative paradigm is its ability to extrapolate far beyond the training horizon. While we train with $T_{\max} = 32$ to promote efficient research strategies, the learned agent can seamlessly operate with $T_{\max} = 2048$ or even higher during inference—a 64× extrapolation factor that would be structurally impossible for mono-contextual approaches.

This extrapolation capability is enabled by two fundamental design choices:

- **Workspace**: Each round's decision depends only on the current reconstructed state $(q, \mathcal{M}_t, \{a_{t-1}, \text{TR}_{t-1}\})$, not on absolute position $t$ or the full trajectory history. This position-agnostic design ensures that the agent's decision-making process remains consistent whether at round 10 or round 1000.

- **Report-based Memory**: The evolving report $\mathcal{M}_t$ provides a scale-invariant representation of research progress. Unlike raw trajectory accumulation, the report's bounded complexity ensures the state distribution remains stable regardless of trajectory length, allowing coherent reasoning at any depth.

We deliberately constrain training to $T_{\max} = 32$ for strategic reasons: (1) it provides sufficient signal for learning effective research strategies while keeping computational costs manageable, and (2) it creates pressure for the agent to develop concise exploration patterns rather than relying on exhaustive search. This constrained training paradoxically enhances extrapolation—by learning to maximize information gain within limited rounds, the agent develops robust strategies that scale gracefully when given additional capacity.

Our experiments (Figure 3) empirically validate this extrapolation capability: agents trained with $T_{\max} = 32$, achieve 42.5% accuracy on BrowseComp when extended to $T_{\max} = 2048$ during inference, compared to only 15.2% with $T_{\max} = 32$. This dramatic improvement demonstrates that the agent effectively utilizes the additional exploration capacity without any degradation in decision quality or coherence.

**Contrast with Mono-contextual Limitations.** Mono-contextual approaches face fundamental barriers to such extreme extrapolation:

- **Position Embedding Overflow**: Absolute position encodings trained on sequences of length 32 often produce undefined or degraded representations beyond the training range

- **Attention Pattern Collapse**: Attention distributions learned on short sequences fail to generalize to dramatically longer contexts, leading to degenerate focus patterns

- **Context Saturation**: The accumulated context from 2048 rounds would exceed most models' context limits, causing hard failures rather than graceful degradation

**Theoretical Foundation.** The extrapolation capability stems directly from our MDP formulation where the optimal policy is defined over states, not trajectory positions. Since our state space $\mathcal{S}$ and decision space $\mathcal{D}$ remain constant regardless of horizon length, a policy learned on shorter trajectories naturally generalizes to longer ones, provided the state distribution remains similar. The report synthesis mechanism ensures this distributional stability by maintaining bounded complexity $O(|\mathcal{M}|)$ regardless of trajectory length, preventing the distribution shift that would otherwise occur with unbounded context accumulation.

This extrapolation capability fundamentally expands the applicability of our approach: agents can be efficiently trained on moderate-length trajectories yet deployed on arbitrarily complex tasks requiring extensive exploration, providing a practical path to handling real-world research challenges of unknown complexity.

### B.4    TRAINING DYNAMICS OF EFFICIENCY-AWARE POLICY OPTIMIZATION

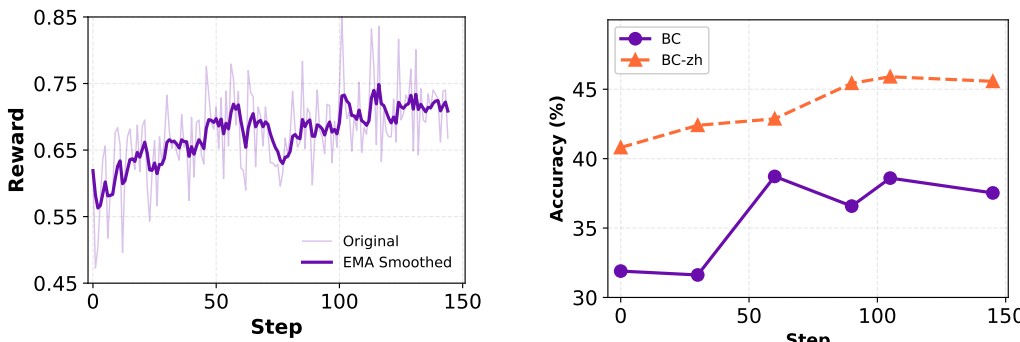

Figure 5: Training dynamics of our RL. **(Left)** Training Rewards Curve. **(Right)** Accuracy Curve.

Figure 5 illustrates the training dynamics of our EAPO framework across 150 optimization steps.

**Reward Convergence.**    The left panel demonstrates stable convergence with training rewards increasing from 0.55 to approximately 0.72, representing a 30.9% improvement. The smooth EMA curve exhibits only minor oscillations, confirming that our adaptive downsampling successfully handles variable sample counts from our iterative paradigm while maintaining stable gradient signals. The consistent upward trend without plateauing suggests the geometric discounting continues to provide meaningful learning signals throughout training.

**Performance Evolution.**    The right panel reveals distinct learning patterns that reflect fundamental differences in task characteristics:

- **BrowseComp (English)**: The sharp performance jump from 32% to 39% at step 50 followed by stabilization suggests the agent discovers critical search strategies—likely effective query reformulation or result filtering patterns specific to English web content. The subsequent plateau indicates these strategies generalize robustly.
- **BrowseComp-zh (Chinese)**: The monotonic improvement from 40% to 45% reflects a smoother optimization landscape, possibly due to more structured Chinese web content or different information organization patterns that allow incremental strategy refinement.

The correlation between reward growth and performance improvement validates our core hypothesis: geometric discounted rewards successfully guide the agent toward more efficient exploration. Notably, the reward improvement (30.9%) exceeds the accuracy gains (BC: 18.8%, BC-zh: 12.5%), indicating the agent learns not just to solve tasks but to solve them *efficiently*. This is empirically confirmed in our ablation studies (Table 2), where EAPO achieves 5.7% shorter trajectories than standard GSPO while maintaining comparable accuracy, demonstrating that our reward design successfully shapes more focused exploration behaviors without compromising task performance.

## C    MORE IMPLEMENTATION DETAILS

In this section, we provide a comprehensive implementation details of our proposed method. For additional insights and more intricate details, we refer the reader to our supplementary materials.

### C.1    ALGORITHMIC FRAMEWORK

Algorithm 1 presents the complete procedure of our iterative deep-research paradigm.

The algorithm proceeds through discrete research rounds, where each round $t$ follows a structured sequence:

---

**Algorithm 1** Iterative Deep-Research (IterResearch)

---

**Require:** Question $q$, Agent model $\pi$, Environment $\mathcal{E}$, Max rounds $T_{\max}$
**Ensure:** Final answer to $q$
 1: Initialize: $\mathcal{M}_0 \leftarrow \emptyset$, $s_0 \leftarrow (q, \mathcal{M}_0, \emptyset)$, $t \leftarrow 0$                    ▷ Empty report and context
 2: **while** $t < T_{\max}$ **do**
 3:      $d_t \leftarrow \pi(s_t)$                                    ▷ Generate structured decision
 4:      Parse: $(\text{Think}_t, \mathcal{M}_{t+1}, a_t) \leftarrow d_t$
 5:      **if** $a_t = \texttt{answer}$ **then**
 6:          **break**                            ▷ Agent decides to terminate
 7:      **end if**
 8:      $\text{TR}_t \leftarrow \mathcal{E}(a_t)$                          ▷ Execute tool and get response
 9:      $s_{t+1} \leftarrow (q, \mathcal{M}_{t+1}, \{a_t, \text{TR}_t\})$           ▷ Reconstruct workspace
10:      $t \leftarrow t + 1$
11: **end while**
12: **return** final answer

---

1. **Decision Generation** (Lines 3-4): The agent $\pi$ processes the current state $s_t$ to produce a structured decision $d_t$, which is then parsed into three components: reasoning ($\text{Think}_t$), updated report ($\mathcal{M}_{t+1}$), and next action ($a_t$).

2. **Termination Check** (Lines 5-7): If the agent outputs $a_t = \texttt{answer}$, the algorithm terminates with the agent's final answer. This allows autonomous determination of information sufficiency.

3. **Tool Execution** (Line 8): For non-terminal actions, the environment $\mathcal{E}$ executes the requested tool (search, browse, compute) and returns the response $\text{TR}_t$.

4. **Workspace Reconstruction** (Line 9): The crucial step distinguishing our paradigm—instead of appending to an ever-growing context, we reconstruct a bounded workspace containing only the question $q$, updated report $\mathcal{M}_{t+1}$, and latest interaction $\{a_t, \text{TR}_t\}$.

**Report Evolution Mechanism.** The report $\mathcal{M}_{t+1}$ serves as the agent's evolving memory, dynamically synthesizing information across rounds. At each step, the agent updates the report by incorporating new findings from $\text{TR}_t$ while preserving essential insights from $\mathcal{M}_t$. This selective retention ensures critical findings persist while redundant information is filtered, maintaining bounded complexity regardless of trajectory length.

**Termination Conditions.** The algorithm terminates under two conditions: (1) **Natural Termination**: The agent determines sufficient information has been gathered and outputs an answer (Line 6). (2) **Forced Termination**: The round counter reaches $T_{\max}$ (Line 11), preventing infinite loops.

## C.2 TOOL ENVIRONMENT

Our environment $\mathcal{E}$ provides four complementary tools that enable comprehensive research capabilities. Each tool is designed to handle specific aspects of the research process, from information gathering to computational analysis. We provide the detailed tool schema in Appendix E.1. We implement the tool environment using production-grade APIs and services:

- **Google Search**: Returns top-10 search results with snippets for general web queries. The tool accepts multiple queries in a single call, enabling efficient batch searching. Each result includes title, URL, and a brief snippet, providing the agent with sufficient context to determine relevance before deeper exploration.

- **Google Scholar**: Returns top-10 search results with snippets for academic papers, citations, and scholarly metadata. Similar to web search, it supports batch queries and returns structured bibliographic information including authors, publication venues, citation counts, and abstract snippets. The tool also includes fallback to general web search for comprehensive coverage. Both Google Search and Google Scholar are accessed via SerpAPI[1], providing reliable and rate-limited access to search results.

---

[1] https://serpapi.com/

- **Visit (Web Browser)**: Enables detailed content extraction from specific URLs with goal-oriented summarization. The agent specifies both the target URLs and a specific goal (e.g., "find the methodology section" or "extract statistical results"), allowing focused information extraction. The tool handles both HTML webpages and PDF documents, automatically detecting and parsing the appropriate format. Our summarization model (Qwen3-30B-A3B) processes the raw content with the agent's goal to produce concise, relevant summaries. We employ Jina Reader[2] for robust web content extraction.

- **Python Interpreter**: Executes arbitrary Python code in a secure, sandboxed environment for computational tasks and data analysis. The interpreter comes with standard libraries (NumPy, Pandas, Matplotlib, etc.) pre-installed and can handle complex calculations, data manipulations, and logical operations. All outputs must be explicitly printed, ensuring clear communication of results back to the agent. We use Code Sandbox[3], ensuring secure and isolated computation.

## C.3 IMPLEMENTATION DETAILS

This section provides comprehensive implementation details of our IterResearch. We also provide all of the code and training data for easy reproduction.

Table 4: Key hyperparameters in the supervised warm-up phase.

| Hyperparameter | Value |
| --- | --- |
| Learning Rate | 1e-5 |
| Batch size | 512 |
| #Epochs | 3 |
| Chat template | `Qwen` (Yang et al., 2025) |
| Maximum Context Length (Prompt + Response) | 40960 |
| Warmup ratio | 0.03 |
| LR scheduler type | Cosine |

**Supervised Fine-tuning Phase.** Since existing LLMs lack inherent capabilities for our iterative deep-research paradigm, we conduct a two-stage data preparation process: **Stage 1: High-quality QA Collection.** We curate 30K high-quality question-answer pairs from recent web research datasets (Li et al., 2025a; Tao et al., 2025; Chen et al., 2025c; Qiao et al., 2025). These pairs are filtered based on answer quality, factual accuracy, and research complexity to ensure they require genuine multi-step investigation. **Stage 2: Trajectory Synthesis.** To bridge the gap between standard QA pairs and our iterative paradigm, we employ Qwen3-235B-A22B (Yang et al., 2025) to synthesize research trajectories following our framework. This process yields 110K training trajectories with an average of 3.7 rounds per trajectory, providing rich supervision for learning the iterative research pattern. We utilize `Slime`[4] as our training framework for the initial supervised fine-tuning phase. The detailed hyper-parameters for this phase are presented in Table 4.

**Reinforcement Learning Phase.** We employ a strategic data selection process to identify questions with optimal learning potential: (1) **Difficulty Calibration**: Using the best checkpoint from SFT, we evaluate each of the 30K questions with 5 independent trials, recording success rates. (2) **Learning Zone Selection**: We retain questions with success rates between 20%-60% (1-3 correct out of 5 attempts), identifying 4,096 questions that fall within the model's "zone of proximal development"—challenging enough to provide learning signal but achievable enough to generate successful trajectories. Questions that are too easy ($> 60\%$ success) provide weak learning signals, while overly difficult questions ($< 20\%$ success) lead to sparse rewards and unstable training. Table 5 summarizes the key hyperparameters used during the reinforcement learning phase. We also use `Slime` as our RL frameowrk due to its efficient and easy to use.

---

[2] https://jina.ai/

[3] https://github.com/bytedance/SandboxFusion

[4] https://github.com/THUDM/slime

Table 5: Key hyperparameters in the RL phase.

| Hyperparameter | Value |
|---|---|
| Learning Rate | 1e-6 |
| Base model | Qwen3-30B-A3B (Yang et al., 2025) |
| Batch size | 16 |
| Group size per Question ($G$) | 16 |
| temperature | 1.0 |
| top p | 0.95 |
| KL loss coefficient ($\lambda$) | 0. |
| entropy coefficient | 0. |
| Maximum Context Length (Prompt + Response) | 40960 |
| Maximum interaction rounds ($T_{\max}$) | 32 |

Table 6: Maximum round settings across different stages and benchmarks.

| Stage/Benchmark | $T_{\max}$ |
|---|---|
| Training | 32 |
| Inference Phase: | |
| GAIA (Mialon et al., 2023) | 32 |
| HLE (Phan et al., 2025) | 64 |
| BrowseComp-zh (Zhou et al., 2025a) | 64 |
| BrowseComp (Wei et al., 2025a) | 256 |

**Maximum Round Settings.** We adopt task-adaptive $T_{\max}$ values to balance training efficiency with inference flexibility. Table 6 summarizes our configuration across different stages and benchmarks. We constrain $T_{\max} = 32$ during both SFT and RL phases to instill efficiency-oriented behaviors. This limit, combined with our geometric reward discounting (Equation 5), creates strong incentives for the agent to develop concise research strategies rather than exhaustive exploration patterns. During inference, we adjust $T_{\max}$ based on benchmark characteristics. This adaptive configuration ensures that simple tasks remain efficient while complex questions have sufficient exploration budget, all while maintaining the efficiency patterns learned during training.

**Reward Design.** We employ LLM-as-judge evaluation following established practices (Chen et al., 2025a). Specifically, we use `Qwen3-235B-A22B` to assess answer correctness:

$$R_T = \begin{cases} 1.0 & \text{if answer is correct} \\ 0.0 & \text{otherwise} \end{cases} \tag{12}$$

**Baselines.** The baseline performance figures for prompting-based comparisons in Table 1 (e.g., ReAct on BrowseComp) were sourced directly from the State-of-the-Art (SOTA) results reported in their original papers. We adopted this methodology for two critical reasons: (1) Many of these prior SOTA works do not publicly release their exact optimized prompts. (2) Therefore, directly reporting their published score is the most rigorous and fair method for comparison. This ensures our results are benchmarked against the true SOTA, not a potentially weaker or un-optimized re-implementation.

### C.4 AVERAGE TOKEN BUDGETS

To provide further insight into the reproducibility and computational cost of our method, we detail the average generation tokens required for the IterResearch-30B-A3B agent to complete tasks across the six primary benchmarks.

It is important to note that this data **excludes** tokens returned by tools (e.g., the content of web pages or search results). It represents only the tokens consumed by the agent's own "Think" and "Report" steps. This figure thus reflects the actual computational cost of the agent's internal reasoning.

Table 7: Average generation tokens (excluding tool responses) for IterResearch-30B-A3B across benchmarks.

| Benchmark | HLE | BC | BC-zh | GAIA | Xbench-DS | Seal-0 |
|---|---|---|---|---|---|---|
| **Avg. Tokens** | 31,426 | 376,349 | 81,136 | 33,198 | 31,600 | 28,186 |

As shown in Table 7, the required reasoning cost (i.e., generation tokens) varies dramatically across benchmarks.

- **BrowseComp (BC)** consumes a token budget (approx. 376k) that is an order of magnitude larger than all other benchmarks. This provides strong empirical evidence of its extreme complexity as a long-horizon task, requiring the agent to perform extensive iteration, thinking, and report synthesis to succeed in the complex web environment. This aligns with our scaling experiments (Figure 3), where performance continually improves as more interaction steps (and thus more generation tokens) are permitted.

- **BrowseComp-zh (BC-zh)**, at approx. 81k tokens, is the second most complex task.

- The other four benchmarks (**GAIA**, **HLE**, **Seal-0**, **Xbench-DS**) are of a similar magnitude (approx. 28k-33k tokens), suggesting that while they are complex multi-step reasoning tasks, they require significantly less iterative synthesis compared to the long-horizon 'BC' benchmarks.

This analysis demonstrates that our method's performance gains (e.g., in Table 2) do not stem from uncontrolled token expenditure. Instead, they are achieved by a superior paradigm (IterResearch) operating within a manageable, task-dependent reasoning budget.

# D  CASE STUDY OF ITERRESEARCH

We present a representative example demonstrating how IterResearch solves a complex biology question through iterative research. This case highlights three key capabilities: (1) evolving report synthesis, (2) efficient information gathering, and (3) autonomous termination decision.

---

**Examples of our IterResearch in Humanity's Last Exam (HLE)**

**data source:** Humanity's Last Exam
**split:** test
**question:** A researcher is interested in studying insects in the family Micromalthidae. They develop a lab colony where larvae are supplied with decaying wood to feed on. Suppose an adult male is found in the colony. Upon its death, what will be the only thing that this individual will have fed on?
Answer Choices: A. Its mother B. Fungus C. Decaying wood D. Cellulose E. Nothing

**Iteration 1**

In the initial iteration, the evolving report and intermediate context are empty.
<**think**>
First, the question is about a researcher studying insects in the family Micromalthidae. They have a lab colony with larvae fed decaying wood. An adult male is found, and upon its death, we need to determine what it fed on based on the answer choices: A. Its mother, B. Fungus, C. Decaying wood, D. Cellulose, E. Nothing.

I need to understand the biology of Micromalthidae. I recall that Micromalthidae is a family of beetles, and they might have unique feeding habits or life cycles. Specifically, I think Micromalthus debilis is a species in this family, and it's known for having a complex life cycle with paedogenesis, where larvae can produce offspring. But the question is about an adult male and what it fed on. Adult males might not feed at all if they have a short lifespan or if their role is solely reproduction. The answer choices include "Its mother," which sounds bizarre, but it might be related to matriphagy or something similar. Also, fungus and decaying wood are common food sources for wood-feeding insects, but cellulose

is a component of wood. I should verify the feeding habits of Micromalthidae adults. Since this is a specific entomological question, I need to search for reliable information.
Available tools include google_search, google_scholar, Visit, and PythonInterpreter. For this, I should use google_scholar or google_search to find academic sources on Micromalthidae feeding behavior.

Let me plan my approach:

1. Search for information on Micromalthidae, specifically about adult feeding habits.

2. Focus on the family Micromalthidae and any unique aspects like matriphagy or non-feeding adults.

I remember that in some insects, adult males don't feed, but I need confirmation for this family. Also, the question mentions "upon its death," which might imply that it didn't feed much or at all, leading to choice E. Nothing. But let's not jump to conclusions. I need evidence. I'll start by searching using google_scholar for Micromalthidae feeding habits or life cycle. Since the question is in English, my output should be in English. Now, for the report section, I need to provide:

- Problem Essence Analysis: Deeply analyze core elements, complexity levels, and potential pitfalls.
- Problem-Solving Planning: Based on understanding, clarify what needs to be confirmed, information sufficiency, and cognitive blind spots.

Then, I must make one tool call.

Potential pitfalls: Misinterpreting the life cycle or feeding habits. Micromalthidae might have unusual behaviors.

Cognitive blind spots: Assuming adult beetles always feed, but some might not.

Information sufficiency: I currently don't know enough, so I need to search. I should search for "Micromalthidae adult feeding" or "Micromalthus debilis life cycle". Let me formulate the search query.

Query: "Micromalthidae adult male feeding" Or more broadly: "Micromalthidae life cycle"

But the question specifies adult male, so perhaps focus on that. Also, "matriphagy in Micromalthidae" because choice A is "Its mother," which might relate to that. I recall that in Micromalthus debilis, there is matriphagy, where larvae eat the mother. But the question is about an adult male, not larva. The adult male might not feed, or perhaps it has a different diet. Upon second thought, in some sources, adult Micromalthus might not feed at all. But let's verify. I'll use google_scholar to search. For the tool call, I need to specify the query. Let me decide on the query. Since google_scholar returns academic papers, I should use precise terms.

Query: "Micromalthidae adult feeding habits" Or "Micromalthus debilis adult nutrition" But to be safe, I'll use multiple queries to cover bases. The tool allows multiple queries, so I can do an array. Let me think. First, I need to confirm the feeding behavior. Also, the Visit tool might be useful if I find specific URLs, but first, I need to search. PythonInterpreter probably isn't needed for this, as it's about information retrieval, not calculation. So, I'll use google_scholar. Now, for the report section. Problem Essence Analysis: The core is to determine the diet of an adult male Micromalthidae insect based on its life cycle and biological characteristics. Complexity arises from potential unique behaviors like matriphagy or non-feeding adults. Pitfalls include misattributing larval behaviors to adults or generalizing from other beetles. Problem-Solving Planning: I need to confirm the feeding

habits of adult males in Micromalthidae. Current knowledge is insufficient, so I must search academic sources. Blind spots might be assuming all adults feed or not, without specific data. Then, tool call: Use google_scholar with queries about Micromalthidae adult feeding. Let me write the queries. Queries: ["Micromalthidae adult male feeding", "Micromalthus debilis life cycle", "matriphagy Micromalthidae"] I included matriphagy because it might be relevant, even though it's about larvae eating mother, but perhaps adults have connections. But the question is about adult male feeding, so focus on that. Perhaps "do Micromalthidae adults feed?" But let's go with the queries.

Now, output format: First <report>, then <tool_call>. Language: English. So, draft the report.
</think>
<report>
### Status Report and Deep Analysis
*Problem Essence Analysis**: The core of this problem lies in understanding the unique biological characteristics and feeding habits of insects in the family Micromalthidae, specifically focusing on adult males. Micromalthidae is a small family of beetles, with Micromalthus debilis being the most studied species, known for its complex life cycle involving paedogenesis (reproduction by larvae) and potential cannibalistic behaviors. The complexity here stems from the possibility of unusual feeding patterns, such as matriphagy (consumption of the mother by offspring) or non-feeding adults, which could influence the answer. Potential pitfalls include conflating larval behaviors with adult ones, misinterpreting ecological roles, or generalizing from other beetle families without specific evidence. The answer choices range from logical options like decaying wood or fungus to more extreme possibilities like matriphagy or complete lack of feeding, requiring precise taxonomic knowledge.

**Problem-Solving Planning**: Based on the current understanding, the key information needed is the confirmed feeding behavior of adult male Micromalthidae insects. While larvae are supplied decaying wood, adults may have different diets or may not feed at all due to short lifespans or specialized roles. Information sufficiency is low without verified sources, as personal recall suggests possible non-feeding or matriphagy-related behaviors, but this must be validated. Cognitive blind spots include assuming all adult beetles feed or overlooking species-specific anomalies. The next step is to search academic literature using Google Scholar to retrieve authoritative sources on Micromalthidae life cycle and feeding ecology, focusing on adult males.
</report>

<tool_call>
"name": "google_scholar", "arguments": "query": ["Micromalthidae adult male feeding", "Micromalthus debilis life cycle", "matriphagy Micromalthidae", "do Micromalthidae adults feed"]
</tool_call>

We will get the tool resposne from Environment here.

<tool_response>
A Google scholar for 'Micromalthidae adult male feeding' found 10 results:
## Scholar Results
1. [5.3 Micromalthidae Barber, 1913](no available link)
publicationInfo: T Hörnschemeyer - Coleoptera, Beetles. Morphology and . . . , 2016 - books.google.com
Date published: 2016
citedBy: 3
. . . larva of cerambycoid type, which is legless and feeds on wood and probably the fungi therein. It . . . From this pupa finally an adult male hatches. Males are haploid with ten chromosomes. . . .
</tool_response>

2. [The remarkable life-history of a new ...
As the content is too long, we will omit it here.
</tool_response>

**Iteration 2**

**Workspace Reconstruction:** Following our transition function, each iteration maintains only: (1) the evolving report between <report>...</report> tags, (2) the current action between <tool_call>...</tool_call> tags, and (3) the previous tool response between <tool_response>...</tool_response> tags. Historical interactions are compressed into the report, not retained in full.

<think>
Hmm, let's start by understanding the question. The user is asking about what an adult male Micromalthidae insect would have fed on before death in a lab colony with decaying wood. The answer choices include its mother, fungus, decaying wood, cellulose, or nothing.

Looking at the previous tool call, I used Google Scholar to search for information on Micromalthidae feeding habits, life cycle, and matriphagy. The results from the tool response are quite detailed. Several sources mention that male larvae practice matriphagy, eating their mothers. For example, the Current Biology article by Normark (2013) states that it takes a male about a week to devour his entire mother before pupating into an adult male. Other papers like Pollock & Normark (2002) and Perotti et al. (2016) also confirm this behavior.

The Wikipedia entry and other web results reinforce this, mentioning that adult males rarely develop and consume their mothers. The key point here is that adult males themselves don't feed; it's the male larvae that engage in matriphagy. Once they become adults, they might not feed at all, as indicated by references to vestigial reproductive organs and non-functional adults.

So, putting it together: the adult male hasn't fed anything after becoming an adult because its feeding happened during the larval stage as part of matriphagy. Therefore, the only thing it fed on was its mother. The correct answer should be A. Its mother.

I need to make sure there's no conflicting info. Some sources mention larvae feeding on decaying wood or fungus, but specifically for males, the matriphagy is highlighted. Also, the question specifies "upon its death," referring to the adult male, so we're looking back at what it consumed during its life. Since the adult doesn't feed, the last thing it ate was its mother during the larval stage.

The status report should document all findings, emphasizing the matriphagy behavior and the fact that adult males don't feed. The next steps aren't needed since the information is sufficient from the sources. So, the final answer is A.

</think>

<report>
### Status Report and Deep Analysis

**Progress Achieved:**

Based on comprehensive research through Google Scholar and web searches, I have gathered substantial information about Micromalthidae biology, specifically focusing on Micromalthus debilis, the only extant species in this family. The key findings are:

1. **Unique Life Cycle Characteristics**: Micromalthus debilis exhibits one of the most complex life cycles among beetles, involving multiple reproductive strategies including

paedogenesis (reproduction by larvae), haplodiploidy, thelytoky, and arrhenotoky. The species is primarily parthenogenetic with female larvae reproducing asexually.

2. **Male Development and Feeding Behavior**: Male development occurs through a specialized pathway where male eggs hatch into active triungulin larvae that immediately begin feeding. Multiple authoritative sources (Normark 2013 in Current Biology, Pollock & Normark 2002, Perotti et al. 2016) confirm that male larvae exhibit obligate matriphagy - they consume their mothers from within. According to Normark (2013): "It takes a male about a week to devour his entire mother, after which he pupates and molts into a winged adult male."

3. **Adult Feeding Habits**: Adult males of Micromalthus debilis appear to be non-feeding or have vestigial mouthparts. Research by Beutel & Hörnschemeyer (2002) and Yavorskaya et al. (2018) indicates that adults have reduced morphological features related to feeding, suggesting they do not feed as adults. The Wikipedia entry on telephone-pole beetles confirms that adult beetles have "vestigial reproductive organs" and likely do not feed.

4. **Contrast with Larval Feeding**: While larvae (particularly female larvae) do feed on decaying wood and associated fungi, this feeding behavior is specific to the larval stage and different between sexes. Male larvae have a specialized feeding requirement focused on matriphagy rather than wood consumption.

5. **Ecological Context**: The laboratory colony described in the question provides decaying wood, but this substrate is primarily utilized by female larvae rather than male individuals. The matriphagy behavior represents an evolutionary adaptation to nutrient acquisition in nutrient-poor environments.

**Source Verification Status**:

- High confidence in Normark (2013) Current Biology article and Pollock & Normark (2002) journal paper

- Supporting evidence from Perotti et al. (2016) Scientific Reports

- Morphological studies (Beutel & Hörnschemeyer 2002, Yavorskaya et al. 2018) provide additional confirmation

- Wikipedia and other secondary sources align with academic literature

**Uncertainties Resolved**:

- Confirmed that adult males do not feed as adults

- Established that male larvae exclusively consume their mothers

- Clarified that decaying wood feeding is primarily associated with female larvae

**Next Steps Plan:**

No further research is required as sufficient authoritative information has been obtained to answer the question definitively. The collected evidence clearly establishes the feeding history of adult male Micromalthidae insects.
</report>

Having synthesized sufficient evidence, the agent autonomously decides to terminate with a comprehensive answer. The evolving report now contains all critical findings, enabling

confident conclusion without further exploration.

<**answer**>

Based on comprehensive research into the biology of Micromalthus debilis, the sole extant species in the family Micromalthidae, the correct answer is A. Its mother.

**Detailed Explanation:**

The telephone-pole beetle (Micromalthus debilis) possesses one of the most extraordinary and complex life cycles known in the insect world, particularly regarding the feeding behavior of male individuals. When an adult male is found in a laboratory colony where larvae are supplied with decaying wood, this individual will have exclusively fed on its mother during its developmental history.

**Matriphagy as Obligatory Feeding Strategy:**

Male Micromalthus debilis develop through a unique pathway characterized by obligate matriphagy - the consumption of the mother by her offspring. This process begins when male eggs are laid by paedogenetic (reproducing while still in larval form) females. These eggs hatch into active, legged triungulin larvae that immediately commence feeding on their living mother from within. According to research published in Current Biology (Normark, 2013), this feeding process requires approximately one week to completely devour the maternal body. Only after consuming sufficient nutrients from their mother do the male larvae pupate and eventually emerge as winged adult males.

**Non-Feeding Adult Stage:**

Critical research by Beutel and Hörnschemeyer (2002) in Systematic Entomology and Yavorskaya et al. (2018) in Arthropod Structure & Development reveals that adult Micromalthus debilis exhibit significant morphological reduction in feeding structures. Their mouthparts are vestigial, and they possess non-functional digestive systems, confirming that adult males do not feed at all during their brief adult stage. The adults represent the terminal phase of the life cycle, having completed all necessary feeding during their larval development.

**Contrast with Female Feeding Behavior:**

While the laboratory colony provides decaying wood as food for the larvae, this substrate is primarily consumed by female larvae rather than males. Female Micromalthus debilis feed on rotting wood, often infested with fungi, which provides them with the energy required for their parthenogenetic reproduction. The male-specific matriphagy strategy represents an evolutionary adaptation to obtain concentrated nutrients in environments where wood decomposition products may be nutritionally limited.

**Evolutionary Context:**

This extreme feeding specialization reflects the unique selective pressures facing Micromalthus debilis. The matriphagy behavior ensures that male offspring receive high-quality nutrients directly from their genetically identical mother (given thelytokious parthenogenesis), while allowing the population to maintain its asexual reproductive strategy. The vestigial nature of adult males suggests they serve primarily as dispersal units rather than feeding organisms.

Therefore, upon the death of an adult male Micromalthidae in the described laboratory colony, the only thing this individual would have consumed throughout its entire life cycle is its mother, making option A the correct answer.

> **</answer>**

# E    INSTRUCTION TEMPLATES

## E.1    TOOL SCHEMA SPECIFICATION

The agent interacts with tools through a structured schema that defines available functions and their parameters. Below we present the complete tool specifications used in our system.

Listing 1: Google Search Tool Schema

```json
{
  "type": "function",
  "function": {
    "name": "google_search",
    "description": "Perform Google web searches then returns a
                    string of the top search results. Accepts
                    multiple queries.",
    "parameters": {
      "type": "object",
      "properties": {
        "query": {
          "type": "array",
          "items": {"type": "string"},
          "minItems": 1,
          "description": "The list of search queries."
        }
      },
      "required": ["query"]
    }
  }
}
```

Listing 2: Google Scholar Tool Schema

```json
{
  "type": "function",
  "function": {
    "name": "google_scholar",
    "description": "Leverage Google Scholar to retrieve relevant
                    information from academic publications. This
                    tool also returns results from Google search.",
    "parameters": {
      "type": "object",
      "properties": {
        "query": {
          "type": "array",
          "items": {"type": "string"},
          "minItems": 1,
          "description": "The list of search queries."
        }
      },
      "required": ["query"]
    }
  }
}
```

Listing 3: Visit (Web Browser) Tool Schema

```json
{
  "type": "function",
```

```
  "function": {
    "name": "Visit",
    "description": "Visit webpage(s) or paper(s) and return
                    the summary of the content.",
    "parameters": {
      "type": "object",
      "properties": {
        "url": {
          "type": "array",
          "items": {"type": "string"},
          "minItems": 1,
          "description": "The URL(s) to visit."
        },
        "goal": {
          "type": "string",
          "description": "The goal of the visit."
        },
        "parse_type": {
          "type": "string",
          "enum": ["html", "pdf"],
          "default": "html",
          "description": "Specify 'html' or 'pdf' format."
        }
      },
      "required": ["url", "goal"]
    }
  }
}
```

Listing 4: Python Interpreter Tool Schema

```
{
  "type": "function",
  "function": {
    "name": "PythonInterpreter",
    "description": "Executes Python code in a secure sandbox.
                    Designed for calculations, data manipulations,
                    and general programming tasks.",
    "parameters": {
      "type": "object",
      "properties": {
        "code": {
          "type": "string",
          "description": "The Python code to execute. Output
                          must use print() functions."
        }
      },
      "required": ["code"]
    }
  }
}
```

## E.2   INSTRUCTION OF OUR ITERRESEARCH

**Prompt of our IterResearch**

You are a professional problem-solving agent with rigorous information verification capabilities and deep analytical thinking.

## CRITICAL OUTPUT FORMAT REQUIREMENTS
You MUST follow this exact format. Every response must contain:

1. <report>...</report> (always required)
2. Either <answer>...</answer> OR <tool_call>...</tool_call> (never both)

## Input Format
- **Current Date**: Current Date
- **Question**: The problem posed by the user that needs to be solved
- **Last Status Report and Deep Analysis**: A summary overview of current work progress
- **Last Tool Call**: The specific action taken in the previous round
- **Last Observation**: The results and feedback obtained after the previous action

## Output Format

<report>
### Status Report and Deep Analysis
**Progress Achieved:**
Based on the Last Status Report and Deep Analysis and Last Tool Response provided in the input, compile a comprehensive and complete documentation of all currently collected information, conclusions, data, and findings. This section must capture ALL important information without any omissions, presented in plain text format with corresponding sources clearly annotated. You must directly record the actual information content rather than using referential markers or summaries. This includes:
1. All factual data and evidence collected
2. All analytical conclusions and insights derived
3. All source materials and their verification status
4. All uncertainties, limitations, or gaps identified
5. Complete integration of previous progress with new findings The documentation must be sufficiently detailed and complete that someone can fully inherit and understand all achieved progress to seamlessly continue the research without losing any critical information or context.

**Next Steps Plan:**
Based on the comprehensive progress achieved above, formulate a detailed and actionable plan for the next phase of research or investigation.
</report> You MUST output this section enclosed with <report></report> tags!

**Decision Point**: Are you certain that no further verification or information gathering is needed to provide the final answer?

**If YES - Information is sufficient:**
<answer>

Answer Format:
1. **Language**: Your answer should be in the same language as the question. If the question uses English, answer in English. If the question uses Chinese, answer in Chinese.
2. The answer should include as much relevant content as possible. Organize the content into separate paragraphs to avoid overly long sections. Avoid content duplication in the answer.
3. Do not include any non-text elements such as URLs, images, or tables that appeared in the reasoning.
4. Output only the answer text. Do not use any additional symbols or start with phrases like 'Here is my answer'.
5. First, output a direct answer to the question.
6. Do not just output the answer to the question; provide a rich and lengthy response by synthesizing all relevant information, and format it using markdown.
7. For statistical data with at least 3 items, use a markdown table to present the results, ensuring the table description is clear. For less than 3 items, describe them directly in text.
8. For research-type questions, try to generate a report of over 1000 words, using subhead-

ings and other elements to improve readability and logic.
</answer>
You MUST output this section enclosed with <answer></answer> tags!

**If NO - Further action needed:**
<tool_call>
"name": "tool name here", "arguments": "parameter name here": parameter value here, "another parameter name here": another parameter value here, ...
</tool_call>
You MUST output this section enclosed with <tool_call></tool_call> tags!

## Working Principles
1. **Rigorous Verification**: Critically evaluate all information sources
2. **Deep Thinking**: Pursue essential understanding, not satisfied with surface phenomena
3. **Evidence-Driven**: Make reasoning decisions based on reliable evidence through deep thinking
4. **You are required to maintain detailed documentation in all your reports and actions, providing sufficient information for others to fully grasp your progress and effectively continue or modify the research trajectory based on your contributions.**

## Special Requirements
- All tools in the tool list are real and functional - as long as you make correct tool calls, you will receive their returned results.
- Clearly distinguish between "confirmed facts," "highly credible inferences," and "hypotheses to be verified"
- Clearly indicate uncertainty when information is insufficient
- Always focus on the original question
- When outputting [Status Report and Deep Analysis], never omit key actions and results, even if these actions or results do not meet expectations, these conclusions must still be documented.
- **When further action is needed, you must select an appropriate tool from your available tool list and carefully configure the tool call parameters based on the tool's specific characteristics and requirements**
- **When the current status is sufficient to answer the question, must provide the final answer enclosed with <answer></answer> tags rather than continue with actions**

## FORMAT REMINDER
- Start with <report>...</report> section
- Then choose: <answer>...</answer> if sufficient info, OR <tool_call>...</tool_call> if need more action
- Never output both answer and tool_call tags in same response

## Input
- Current Date: {date_to_use}
- Question: {question}
- Available Tools
{tools}
- Last Status Report and Deep Analysis:
<report>
{report}
</report>
- Last Tool Call:
<tool_call>
{action}
</tool_call>
- Last Tool Response:

<tool_response>
{observation}
</tool_response>

Now please begin your deep analytical work. The language of your output must be consistent with the language of the question. If the question is in Chinese, output in Chinese; if the question is in English, output in English.

