# OpenReview forum: "IterResearch: Rethinking Long-Horizon Agents with Interaction Scaling"
_ICLR.cc/2026/Conference — ICLR 2026 Poster_

### Official Review · Reviewer_XKsG · 2025-10-30

**Soundness:** 2
**Presentation:** 3
**Contribution:** 2
**Rating:** 4
**Confidence:** 4

**Summary:**

The paper proposes IterResearch, a new method for deep research. Current approaches keep expanding the context in the think->act phases which can limit the space available for a response. IterResearch is inspired from MDPs and compresses the current context by summarizing it into a report. Thus, the new context consists of the original question, the report, and any actions to take.

The authors also introduce EAPO - Efficiency Aware Policy Optimization by using discounted rewards for binary, sparse deep research settings where the reward is only given on the final step.

Finally, IterResearch can also be used as a prompting paradigm directly so that no training etc is needed and can work with current models.

The authors then conduct an empirical evaluation with different baselines.

**Strengths:**

1. The paper is clear and well-written

2. The overall idea is intuitive

3. Results demonstrate significant improvement (although there are some open questions here)

**Weaknesses:**

Overall, the results clearly seem to advance the SOTA so Im hoping for more clarifications to my identified weaknesses. Im happy to engage in discussion and revise my score if my concerns are adequately addressed.

1. I think one of the major claims of the paper is the Markovian State Reconstruction. There is not really any guarantee that this report is Markovian in any sense. The Markovian State suggests that the future state is independent of the history given the current state.

If that were the case, then definitely the transition function with the input history or with the report is not going to change the outcome. So the paper title feels a bit bold. It is more apt to call it summarization but calling it Markovian State Reconstruction is a bit too much. I think experiments would be needed to back your claims.

The idea is intuitive to improve the performance due to limitations on attention windows/distributions but to claim that summarization of the past context and making it a report is making it Markovian seems a bit overblown IMO.

2. I tihnk one problem with the results is that the prompts are not provided for the baselines in the prompt-based approach. The prompt used in appendix F.2 is quite exhaustive and Im not sure if some instructions that are agnostic to the approach were used in the baselines.  For example, the entire section of ## Working Principles. This might be an issue with clarity in the paper.

3. I think the claim of unbounded interactions is overstated. Even summarization has limits. Are the summaries never going to grow? How does the LLM know what information is important. There could be some tool response that is only going to be needed several steps later. This is often the case in partial-order planning. One issue here is that once that information is not included in the summary, it is lost forever. At least in the mono-context case it is possible to be "re-"discovered by a later step based on the generated output and attended to. I am a bit confused on how the LLM knows to generate a good report so that such issues do not occur.

3. I dont think EAPO is really novel. RL algorithms use discounting by definition and this does not seem like a new contribution. Please justify why you think EAPO is novel?

**Questions:**

Please see weaknesses

---

> ### Author Response · Authors · 2025-11-20
> **Response Part 1**
>
> We sincerely thank Reviewer XKsG for the thorough review and the encouraging assessment that our "results clearly seem to advance the SOTA." We appreciate the opportunity to clarify our theoretical framing.
>
> We have addressed your identified weaknesses below and have revised our paper (e.g., Section 3.1.1 and Appendix D.3) to incorporate these clarifications.
>
> > Response to W1:
>
> We appreciate the reviewer’s careful scrutiny of our terminology.
>
> 1. **Formal Guarantee (Refer to General Response 2)**:
> As detailed in General Response 2, our system is Markovian by construction.
> We explicitly define the state transition as $\mathcal{T}(s_{t+1} | s_t, d_t)$, where the policy $\pi(d_t|s_t)$ is architecturally restricted to access only the current workspace $s_t$ (Query + Report + Current Context). Because the policy cannot access the discarded history $h_{<t}$, the future evolution of the system depends mathematically only on the current state $s_t$. Therefore, the Markov property holds by definition of the system's architecture.
>
> 2. **Addressing the "Guarantee" of Information Sufficiency**:
> The reviewer correctly asks: what guarantees that this Report contains necessary information? While the architecture guarantees the Markov property, the sufficiency of the state is guaranteed by **End-to-End Reinforcement Learning optimization**.
>
>     * If the Report fails to capture critical historical information (i.e., it becomes an insufficient statistic), the agent will *fail the task and receive zero reward*.
>     * Through RL training, the agent is explicitly incentivized to construct $s_t$ such that it maximizes $\pi(\text{success}|s_t)$.
>     * **Empirical Proof**: Figure 3 shows performance scaling up to 2048 steps. If the state were "losing" critical information (i.e., failing to function as a valid Markov state), performance would collapse. The continuous improvement proves the agent successfully learns to construct a valid state.
>
> 3. **"Reconstruction" vs. "Summarization"**:
> We chose "Reconstruction" over "Summarization" because our "synthesis" step is not passive compression. The agent is *actively and goal-directedly synthesizing information* relative to the original query $Q$ to build a new state object that answers, "What do we know now about $Q$?" and "What is the plan next?" We believe this active, intelligent synthesis is more accurately described as "reconstruction."
>
> We hope this explanation clarifies our theoretical framing. We thank the reviewer again for prompting this crucial clarification.
>
> > Response to W2:
>
> We apologize for the confusion and have added the baseline details to Appendix D.3. We clarify two critical points:
>
> * **Paradigm vs. Prompting Tricks**: The instructions in Appendix F.2 (e.g., ## Working Principles) are not auxiliary "tricks" but the textual implementation of the IterResearch paradigm itself. They define the iterative Think -> Report -> Action structure. The +19.2pp gain (Fig. 4) stems from this structural shift in reasoning topology, not from specific wording enhancements.
>
> * **Comparison against Standard SOTA**: To ensure the most rigorous and fair comparison, the results for baselines (OpenAI DeepResearch, various Agent frameworks) were sourced directly from their original SOTA papers. This standard practice avoids potential bias from sub-optimal reproduction and ensures we are comparing against the community's best reported numbers.

---

> ### Author Response · Authors · 2025-11-20
> **Response Part 2**
>
> > Response to W3:
>
> We thank the reviewer for this critical question.
>
> 1. "Unbounded" (Theoretical) vs. 2048-Step Scaling (Empirical):
> The reviewer is correct that "unbounded" is a theoretical descriptor, which is precisely how we have framed it in our original paper (e.g., lines 96, 233, 799, 802, Table 3). Our work models the theoretical capability for unbounded interaction because our paradigm fundamentally breaks the O(t) linear dependency between interaction steps and context size.
>
>     Our *practical contribution*, Figure 3, is the **first** empirical demonstration of this theoretical model's implications. We show our agent successfully scaling to 2048 interactions within a only 40K context—with performance continuously improving—which is far beyond the hard limits of any mono-context agent and a feat no other work has yet demonstrated.
>
> 2. The Misunderstanding of "Re-Discovery" in Mono-Context Agents:
> The reviewer suggests that mono-context agents have an advantage because lost information can be "re-discovered." We must respectfully disagree with this premise.
>
>     While information is theoretically present in a mono-context, it is functionally lost due to catastrophic "noise contamination" and the "needle-in-a-haystack" problem. As demonstrated by extensive prior work on long-context retrieval, an agent's ability to find and attend to a critical piece of information plummets as the context grows. The mono-context agent's failure mode is not "loss" but "drowning."
>
> 3. How Our Agent Learns (The RL Guarantee):
> The reviewer's central question is, "How does the LLM know what information is important?" This is the exact problem our agent is trained to solve via Reinforcement Learning (EAPO).
>
>     * The agent is not performing naive, stateless "summarization." It is learning a state reconstruction policy by End-to-End RL.
>
>     * RL provides the guarantee. If the agent deletes critical information (e.g., a "partial-order planning" dependency) that is needed steps later, the final task will fail. This entire trajectory receives a zero reward (Eq. 3).
>
>     * This reward signal is precisely what teaches the agent to distinguish between "noise" (to be discarded) and "essential findings" (to be preserved in the report) relative to the original query $Q$.
>
> 4. The Empirical Proof (The "What"):
> Figure 3 is the definitive experimental answer. **If our agent were randomly or naively losing critical information, its performance would inevitably collapse as interactions grow**.
>
>     Instead, our results show the exact opposite: performance continuously improves (from 3.5% to 42.5%) as the agent scales to 2048 interactions within a fixed 40K context. To our knowledge, no other work has demonstrated this level of scaling. This proves that our agent is successfully learning to manage this trade-off and retain goal-critical information far more effectively than any mono-contextual approach.
>
>
> > Response to W4:
>
>
> As framed in General Response 1, our primary contribution is the IterResearch paradigm. EAPO is presented **as the essential engineering enabler tailored to this paradigm**.
>
> We must respectfully disagree that EAPO is "just standard discounting." The reviewer is correct that RL algorithms use discounting (e.g., $\gamma$), but EAPO's novelty lies in how and why it is used—to solve problems unique to our paradigm:
>
> 1. EAPO is for Efficiency, not just Temporal Credit:
> Standard RL discounting is used for temporal credit assignment. Our reward $r_t = \gamma^{T-t} R_T$ (Eq. 3) is a **specific reward-shaping mechanism** for efficiency. In our sparse, binary-success setting ($R_T \in \{0, 1\}$), this signal explicitly teaches the agent to prefer shorter, correct solutions over longer ones. This "interaction-agnostic" efficiency signal is a non-trivial adaptation for deep research tasks.
>
> 2. EAPO Solves Multi-Round Data Structure (Eq. 4):
> Furthermore, EAPO's novelty is not just Eq. 3. As described in Sec 3.2.2, standard RL algorithms are not designed for our paradigm's unique data structure (where each round is a sample). EAPO's adaptive downsampling (Eq. 4) is a critical, non-standard component that manages the high variance from easy negative samples, enabling stable training.
>
> In summary, EAPO is the necessary, non-trivial engineering to solve the specific training challenges (efficiency, data structure) created by our novel IterResearch paradigm. Providing the "engineering" to make a novel, SOTA-advancing paradigm (IterResearch) practical and trainable is, we believe, a significant and valuable contribution to the community.
>
> ---
>
> We thank the reviewer again for their constructive feedback. We hope these clarifications, and the corresponding revisions to our paper, have fully addressed your concerns. We respectfully hope you will consider these points in your final assessment.

---

> ### Comment · Reviewer_XKsG · 2025-11-25
> **Notes on Response Part 1**
>
> Thank you for your response.
>
> I maintain my stance that the claim of Markovian is a bit overblown. Let me provide some explanation for my thought process. Happy to retract my statement if more clarification is provided.
>
> I took a look at your edits and remain unconvinced about this claim.
> Any system (even a random state evolution system) that has the distribution $P(s_{t+1}|s_t, a_t)$ is Markovian by your definition although the Markov property states that the next state is independent of the history.
>
> As a result, to claim something as Markovian, one needs to show that $P(s_{t+1}|s_t, a_t, s_{t-1}, a_{t-1}, \ldots, s_0) = P(s_{t+1}|s_t, a_t)$. Without showing this, any one can model some random function $s_{t+1}: s_t \times a_t$ and call it markovian. I dont believe this experiment is done and thus i cannot accept that your formulation is truly Markovian even though you have written it formally. You would need to show that the way the report evolves is truly independent of providing it all past previous reports and only then can call it Markovian. This might seem a bit pedantic but the paper text makes it seem that this is a truly Markovian reduction. Either change the text to clarify this or showcase it via experiments.

---

> > ### Author Response · Authors · 2025-11-26
> >
> > We sincerely thank you for the prompt response and for acknowledging the validity of our modeling approach itself. We sincerely apologize for the confusion in the original wording and appreciate the rigor you are applying to the theoretical definition.
> >
> > 1.  Agreement on Mathematical Definition:
> > We fully agree with your rigorous definition. Strictly claiming a state is "Markovian" implies proving it is a sufficient statistic of the history:
> >
> > $$P(s_{t+1}\mid s_t, a_t, s_{t-1},\ldots,s_0)=P(s_{t+1}\mid s_t, a_t).$$
> >
> > Our intention was not to claim that our reduction automatically holds this mathematical property inherently. Instead, our goal is to *formulate* the iterative research procedure as an MDP by *designing* the state representation such that it contains all task-relevant information needed for future decisions. Specifically, we define the reconstructed state as
> >
> > $$
> > s_t = (q, M_t, \{a_{t-1}, TR_{t-1}\}),
> > $$
> >
> > where the evolving report $M_t$ is a compressed summary of past findings $s_{t-1}$, and $\{a_{t-1}, TR_{t-1}\}$ captures the most recent interaction with the environment. Under this \emph{modeling choice}, the agent's transition dynamics depend only on the current state. Our intention is to emphasize that the Markovian property is a feature of our \emph{state design}, rather than an inherent property of the external environment.
> >
> > 2. The challenge of theoretical proof lies in LLM field.
> > However, as you implicitly noted, **rigorously proving that a natural language summary functions as a mathematical 'sufficient statistic' is a widely recognized challenge in the LLM/NLP field**. Given the effectively infinite state space of natural language, providing such theoretical guarantees is generally considered infeasible and is not a standard requirement for empirical agentic frameworks.
> >
> > 3. Clarification of Intent & Shift to Scaling.
> > We acknowledge that our original description may have caused a misunderstanding: we intended to **borrow the architectural intuition of MDPs** to enable **interaction scaling**, rather than claiming to have mathematically proved the sufficient statistic property. As you correctly identified, the true value of this design is Interaction Scaling, allowing agents to operate efficiently over thousands of steps (as shown in Figure 3).
> >
> > 4. Actionable Revision Plan.
> >
> > To strictly align our manuscript with the architectural clarification above and avoid any "overblown" theoretical implications, we have implemented the following revisions:
> >
> > * **Title Change**: We have changed the paper title to **"IterResearch: Rethinking Long-Horizon Agents with Interaction Scaling"**. This explicitly shifts the emphasis from the theoretical "Markovian" claim to our primary empirical contribution.
> >
> > * **Reframing Formulation (Section 3.1.1)**: We have revised the text to emphasize the structural design rather than the property. The revised text reads:
> >
> >     ``This formulation ensures that the transition to $s_{t+1}$ depends solely on the current reconstructed workspace $(s_t, a_t)$, strictly adhering to the MDP intuition to prevent context bloat. The state independence arises from our architectural constraint rather than from assumptions about the external environment.''
> >
> > * Global Terminology Adjustment: We have conducted a full-text review to replace absolute terms like "Markovian state" with more precise engineering descriptions such as "Iterative state" or "Synthesized representation."
> >
> > We hope this clarification resolves the misunderstanding and accurately reflects our intended formulation.

---

> ### Comment · Reviewer_XKsG · 2025-11-25
> **Notes on Response Part 2**
>
> > Our reward  (Eq. 3) is a specific reward-shaping mechanism for efficiency.
>
> This is actually standard practice for representing episodic rewards in RL. Please seen Sutton and Barto 2nd edition, Eqn 3.2 for more details. It does not matter whether rewards are binary or dense in this equation since it represents an episodic reward.
>
> The paper writing makes it seem like a new contribution perhaps, so maybe edits there are needed.
>
> Re
> > The Misunderstanding of "Re-Discovery" in Mono-Context Agents:
>
> Perhaps I was unclear. While current SOTA models do seem to have that problem where they have a hard time getting the info picked, the fundamental thing is that the information is not yet lost. With a report, if the information is lost, it is truly lost and can thus never be rediscovered esp if the tool only was oging to be called  once. Future advances in models could actually recover this information in a mono-context since it is not deleted. The problem here is that once information is lost it is lost forever in IterResearch in contrast to mono-context, where it is not lost but rather currently has a hard time being attended to.

---

> > ### Author Response · Authors · 2025-11-26
> >
> > > Q1 Claim of Reward.
> >
> > Thank you for pointing out the reference to Sutton and Barto. We have checked the text, and we believe that you are referring to Eqn.~3.2 regarding the expected discounted return. You are absolutely right that the discounted reward design is standard in RL. Our intention was not to present discounting itself as a novel contribution, but rather to highlight its specific role within IterResearch.
> >
> > In our framework, the discount factor $\gamma^{T-t}$ naturally emphasizes states closer to the final answer, which aligns well with our decomposition of trajectories into many per-round samples and promotes efficiency during training. Our EAPO represents the combination of this standard efficiency-shaping mechanism with our adaptive downsampling, which is necessary to stabilize training given our paradigm's unique "per-round sample" structure. To avoid misunderstanding, we have revised the text to explicitly clarify this point:
> >
> > ``We apply the standard discounted reward within the iterative decomposition of IterResearch, which enables efficient training by weighting states according to their proximity to the final answer and by decomposing trajectories into multiple per-round samples.''
> >
> >
> > > Q2 Claim of Re-Discovery in Mono-Context Agents.
> >
> > Thank you for the helpful clarification. We agree with your point that mono-context agents do not delete past information; rather, they often struggle to re-attend to earlier content due to attention degradation. In contrast, *IterResearch* performs an intentional and irreversible compression into the report $M_t$, meaning that information outside the report is genuinely discarded, but this design enables substantially more interaction rounds. While mono-context agents typically operate within a few hundred effective steps, our framework supports thousands of interactions, which leads to improved performance in our simulations.
> >
> > We respectfully argue that in the reality of long-horizon research, sacrificing exploration depth is a greater risk than compressing history.
> >
> > 1. Trade-off: Preserved History vs. Sacrificed Exploration.
> >     While Mono-context agents theoretically preserve all historical information, they are **structurally forced to sacrifice exploration depth**.
> >     * **The "Ceiling" Problem**: As you noted, Mono-context agents typically exhaust their context window within ~100 turns. This creates a hard ceiling: any critical information that requires 500 or 1000 steps to uncover is effectively inaccessible to them.
> >     * **Inaccessible Information**: This means any critical information residing deeper in the search space (e.g., at Step 2000) is effectively lost to a Mono-context agent simply because it can never reach that depth. **We argue that the capability to reach the information is a prerequisite for using it**.
> >
> > 2. Physical Limits and Asymptotic Behavior:
> >     Even with stronger future models, the Context Limit imposes a hard physical upper bound (Memory & Inference Cost).
> >    * Asymptotic Failure: As the horizon $T \to \infty$, any linear accumulation paradigm inevitably faces system collapse ("Context Suffocation").
> >    * Computational Necessity: Therefore, "Selective Forgetting" is not just a design choice but a computational necessity for true interaction scaling. Our paradigm provides the architectural solution to this physical constraint.
> >
> > 3. **RL-Driven Safe Compression**:
> >     Crucially, our agent does not perform random forgetting.
> >     * The agent is incentivized to learn an "Optimal Compression Policy."
> >     * If the agent deletes critical information that is needed for the final answer, it receives Zero Reward.
> >     * Consequently, the agent learns to retain the necessary information for the task, ensuring that only noise is discarded while the signal is retained.
> >
> > 4. Empirical Proof
> >     Figure 3 provides the definitive empirical answer. If critical information were being irreversibly lost, performance would degrade as interactions increase. Instead, accuracy **improves dramatically** (3.5% $\to$ 42.5%) as we scale to 2048 turns. This proves the agent successfully learns to retain necessary information even under extreme compression pressure.
> >
> >
> > We genuinely value the rigor you brought to the review process. Your feedback has been instrumental in refining the manuscript's precision—shifting the narrative from "theoretical claims" to "architectural innovations" for interaction scaling. We have uploaded the revised manuscript reflecting all these changes (Title, Abstract, Sections 1, 3, and EAPO citations). We believe the paper now offers a much clearer and more scientifically grounded presentation of our contributions, and we respectfully hope these clarifications and revisions fully address your concerns.

---

> ### Comment · Reviewer_XKsG · 2025-11-26
>
> Thanks for your response and make the the clarification edits to improve the clarity of the paper.
>
> Re
> > We respectfully argue that in the reality of long-horizon research, sacrificing exploration depth is a greater risk than compressing history.
>
> I agree with your follow up points on this.
>
> The rebuttal and follow up have cleared all my concerns now.
>
> As i noted in my review, the results do seem to clearly advance SOTA and the overall approach of using an MDP formulation for the approach is intuitive.
>
> I will raise my score in favor of accepting the work. My only comment is to make the true novelty of EAPO stand out more with another writing pass of the text. I think you can probably move it to an appendix too and perform more ablation studies or provide more intuition on the MDP formulation with the saved space.
>
> In case the paper gets rejected, please keep polishing and resubmit to another venue like ICML.

---

> > ### Author Response · Authors · 2025-11-27
> > **Thank You for the Constructive Dialogue and Score Increase**
> >
> > We are sincerely grateful for your engagement throughout this discussion. We are thrilled to hear that our clarifications regarding the trade-off between exploration depth and history compression have resolved your concerns, and we deeply appreciate your decision to raise the score.
> >
> > Acknowledgment of Suggestions: We are glad to have reached a consensus on the value of the MDP-based formulation for scaling long-horizon research. We also highly value your final suggestion regarding the presentation of EAPO. We agree that refining its description to better highlight its role as an engineering adaptation—and optimizing the space allocation—will further strengthen the manuscript's focus. We carefully incorporate these insights into the revised version to ensure the core contribution of interaction scaling stands out clearly.
> >
> > Thank you again for your time, rigor, and encouraging feedback. Your review has significantly improved the quality of this work. We are committed to incorporating all these changes to ensure the final manuscript meets the highest standard.

---

### Official Review · Reviewer_wSFE · 2025-11-01

**Soundness:** 2
**Presentation:** 3
**Contribution:** 2
**Rating:** 4
**Confidence:** 5

**Summary:**

The paper proposes IterResearch, a long-horizon reasoning framework where an agent maintains a Markovian workspace instead of accumulating the full dialogue history. The evolving report acts as compressed memory, keeping context size nominally constant. Training uses Efficiency-Aware Policy Optimization (EAPO) with geometrically discounted rewards to promote concise, successful trajectories.

IterResearch achieves notable gains (+14.5 pp on six benchmarks) and reportedly scales from 32 to 2048 reasoning steps, outperforming prior open-source systems and even rivaling proprietary ones. However, the "constant-context" claim is partly theoretical - since the report must still grow or overwrite information, true unbounded reasoning is unproven. The paper also lacks experiments confirming recovery from early false summaries or testing smaller training horizons. Comparisons rely on older baselines (GPT-4.1, o4-mini), and it remains unclear whether gains stem from the paradigm itself or its prompting format.

Overall, IterResearch is an elegant compression-based reformulation of long-horizon reasoning, but its claimed scalability and memory efficiency need stronger empirical validation.

**Strengths:**

1. Clear, principled formulation. The Markovian workspace reconstruction is simple and well-motivated, giving constant context size and avoiding "context suffocation"/"noise contamination". The formal transition s_{t+1}=T(s_t,d_t,E(a_t)) is explicit.

2. Strong empirical coverage. Six benchmarks with diverse characteristics; competitive vs. proprietary systems on several, e.g., surpassing OpenAI DeepResearch on HLE and BrowseComp-zh.

3. General usefulness as a prompting recipe. IterResearch prompting outperforms ReAct with frontier models (o3, DeepSeek-V3.1), suggesting paradigm-level value beyond one implementation.

**Weaknesses:**

1. Markovian constant context is a conceptual simplification rather than a true breakthrough.
The claimed $O(1)$ workspace complexity is only formal when the evolving report $|M|$ is treated as fixed. In practice, $|M|$ grows with the number of synthesized summaries, and each tool response $|TR|$ can vary widely across steps. Therefore, the overall process still depends on in memory and computational complexity, merely folded into a different structure. This makes the "constant" context claim more of a heuristic compression scheme than a provably unbounded process. Once the report becomes saturated, the system risks discarding fundamental early-stage information, effectively re-introducing the limitations it sought to remove.

2. Potential inefficiency from report initialization.
If the evolving report $|M|$ is kept constant, the first few interactions still include a large workspace prompt even when the agent has accumulated little information. This may yield unnecessary compute overhead at the start of every episode, contradicting the claimed "efficiency-aware" principle.

3. Questionable extrapolation to long horizons.
The claim of scaling from $T_{max}=32$ (training) to $2048$ interactions (inference)rests on the assumption that the agent can reconstruct forgotten information through new searches. Yet, no evidence is given that this process retrieves previously lost or overwritten insights, as opposed to repeatedly circling around similar queries or rediscovering the same partial results. Without qualitative trajectory inspection, it is unclear whether high-turn runs reflect genuine continued reasoning or redundant loops.

4. No measurements for smaller training horizons.
While Section 4.4 evaluates scaling at inference, there are no experiments showing performance under smaller $T_{max}$ during training. Thus, it is impossible to assess whether longer training horizons produce better generalization or if the model’s extrapolation is accidental.

5. Outdated or inconsistent baseline comparisons.
The benchmark comparison includes GPT-4.1 and o4-mini but omits GPT-5-series or contemporary baselines. This limits the credibility of claims about competitiveness "with frontier proprietary systems".

6. Lack of ablation on report-based prompting alone.
The paper does not include a variant where the same prompt structure is given to a baseline model without the workspace-reconstruction mechanism (i.e., the same input text but no internal state machine). It remains unclear how much of the improvement arises from the report-prompt structure rather than the Markovian update itself.

7. Novelty of EAPO may feel incremental. Geometric discounting for terminal rewards is standard MDP practice; while appropriate here, the RL contribution beyond applying discounting + GSPO + downsampling reads as engineering rather than fundamentally new learning theory. Ablations isolate length reduction but not a deeper analysis of policy changes.

**Questions:**

1. What happens when $T_{max}$ during training is reduced to, say, $8$ or $16$? Does extrapolation still hold, or does generalization degrade proportionally?

2. How can we confirm that high-turn (>100) trajectories generate genuinely new insights rather than re-searching similar content? Have the authors analyzed overlap of retrieved URLs or paraphrased content between early and late rounds?

3. How do accuracy and average step count vary with $\gamma$ (e.g., $0.99$/$0.997$/$0.999$)? Is the $5.7%$ length reduction robust, and where does performance begin to trade off?

4. If the evolving report contains an early false hypothesis, can the agent recover? Have the authors measured recovery rate with stress tests where earlier synthesized content is wrong/misleading?

---

> ### Author Response · Authors · 2025-11-20
> **Response Part 1**
>
> We sincerely thank Reviewer wSFE for the detailed review and for recognizing the "notable gains," "strong empirical coverage," and the "elegant compression-based reformulation."
>
> We have addressed the central concern regarding the definition of our paradigm and the Markov property in our General Response, which serves as the foundation for our replies below.
>
> > Response to W1
>
> We thank the reviewer for this nuanced critique. We agree that $|\mathcal{M}|$ is not rigidly fixed in size, but we clarify the distinct advantage of our approach:
> 1. **Decoupling Context from Time ($t$)**: Our "constant context" claim refers to the computational complexity being **independent of the number of interaction steps $t$ (i.e., $O(1)$ relative to $t$)**. In contrast, mono-contextual agents suffer from $O(t)$ linear growth, guaranteeing failure as $t \to \infty$. Our paradigm decouples memory cost from trajectory length, shifting from a structural bottleneck ($t$) to the information density of the report ($|\mathcal{M}|$), which is the key innovation enabling scaling.
> 2. **Learned Sufficient Statistic vs. Noise Accumulation**: The reviewer characterizes the report as a "heuristic compression scheme." We agree, but emphasize that it is an **optimization-driven heuristic**, not an ad-hoc rule. As detailed in our **General Response**, the report $\mathcal{M}_t$ functions as **an explicit, interpretable, and learnable sufficient statistic**. Through end-to-end RL, the agent learns the optimal trade-off between compression and retention11. This is far superior to the "perfect retention" of mono-contextual agents, which paradoxically leads to "context suffocation" via irreversible noise accumulation.
> 3. **Empirical Validation**: The fear that the system "risks discarding fundamental information" is theoretically valid but empirically disproven by Figure 3. If critical information were being discarded, performance would degrade at long horizons. Instead, performance improves monotonically (3.5% $\to$ 42.5%) up to 2048 steps, proving the learned compression is highly effective.
>
> > Response to W2
>
> This concern stems from a misunderstanding of our initialization and the definition of "constant context."
>
> 1. **The Report Starts Empty**: The reviewer posits that the workspace is "large" at the start. However, as explicitly specified in Eq. 1 and Algorithm 1 (Line 1), the process begins with $s_0 = (q, \mathcal{M}_0, \emptyset)$ where $\mathcal{M}_0 \leftarrow \emptyset$. Thus, the "unnecessary compute overhead" at the start does not exist.
>
> 2. **Clarification on "Constant"**: We clarify that "constant context" does not imply a fixed-size memory block allocated from the start. It means the context size is **structurally decoupled from the interaction step $t$**. While the report $\mathcal{M}_t$ grows organically with the research findings, it does not carry the linear overhead of raw history accumulation ($O(t)$) that plagues baselines. Our approach is inherently efficient, computing attention only on the synthesized signal ($\mathcal{M}_t$) rather than the entire raw history.
>
> > Response to W3 & Q2
>
> The reviewer asks if high-turn runs reflect "genuine continued reasoning" or "redundant loops."
> 1. **Quantitative Proof (Figure 3)**: If the agent were entering redundant loops, performance would plateau or degrade as $T_{max}$ increases. Instead, we observe a **+39pp performance gain** when scaling $T_{max}$ from 2 to 2048. This monotonic improvement proves the agent utilizes the additional budget to solve harder problems, not to loop.
> 2. **Qualitative Proof**: We have added detailed case studies of long-horizon trajectories (100+ steps) to** the supplementary material**. These show the agent autonomously formulating sub-questions, verifying hypotheses, and updating $\mathcal{M}_t$ without cyclic behavior. The "evolving report" acts as a negative constraint against redundancy: since the report explicitly states what is known, the agent learns not to search for it again.

---

> ### Author Response · Authors · 2025-11-20
> **Response Part 2**
>
> > Response to W4 & Q1
>
> We address W4 and Q1 together. This questioning reveals a core misunderstanding of why our extrapolation works. The reviewer asks about training with $T_{max}=8$ or $16$.
>
> We chose $T_{max}=32$ 3 to **ensure a sufficient exploration budget for our training data**. We target "Deep Research" tasks and difficult training data which typically require multi-step reasoning. If we train with $T_{max}=8$, complex trajectories are cut off before success, resulting in sparse rewards and a failure to learn valid research patterns.
> The extrapolation capability is not a "learned artifact" of the number 32; it is a **structural property of the Markovian paradigm**. Because the policy $\pi(s_t)$ is position-agnostic, a behavior learned at $t=30$ (e.g., "summarize and verify") applies equally at $t=2000$.
>
> As long as $T_{max}$ is sufficient for the agent to learn the core research behavior in training data, the paradigm itself guarantees scalability. In summary, the extrapolation is not "accidental"; it is the intended, designed-for consequence of our IterResearch.
>
> > Response to W5
>
> We appreciate this suggestion. We **have updated Table 1** to include the newly released GPT-5 as a high-performance reference. We note that our work was completed in July, prior to GPT-5's release in August. While GPT-5 naturally outperforms our smaller model due to its massive parameter scale, it is notable that our IterResearch (30B-A3B)—despite being orders of magnitude smaller—still achieves performance comparable to, and on some benchmarks surpassing, specialized proprietary systems like OpenAI Deep Research (as shown in Table 1). This highlights the efficiency and effectiveness of our iterative paradigm.
>
> > Response to Weakness 6
>
> The reviewer suggests an ablation: "same prompt structure... but no internal state machine." We respectfully point out that **this ablation describes the Mono-Contextual (ReAct) Baseline**.
> * If we remove the "state machine" (the mechanism that updates/overwrites $\mathcal{M}_t$), we must retain the history to prevent amnesia.
> * Retaining the history *is* the Mono-Contextual paradigm.
> * Therefore, the comparison in Figure 4 (IterResearch vs. ReAct) and Table 2 (IterResearch vs. Mono-Agent)  is exactly this ablatio. The +12.6% gain (Table 2) quantifies the value of the state machine mechanism over the standard prompt structure.
>
>
> > Response to Weakness 7
>
> We appreciate the reviewer’s assessment and agree. EAPO is proposed as **an engineering solution**. As detailed in **General Response 1,** standard RL (GRPO) fails in our paradigm due to (1) lack of efficiency incentives for variable-length research and (2) batch instability from trajectory decomposition. **EAPO is the necessary engineering adaptation that makes the novel IterResearch paradigm trainable**.
>
> > Response to Q3
>
> We thank the reviewer for this insightful question regarding hyperparameter sensitivity. To address this, we present our detailed experimental results comparing different $\gamma$ values across all six benchmarks.
>
> 1. Empirical Sensitivity Analysis:
> During our development phase, we performed a sensitivity sweep on the discount factor $\gamma \in [0.99, 0.999]$. As shown in the table below, we observe a clear "sweet spot" behavior.
>
> | $\gamma$ Value | HLE | BC | BC-zh | GAIA | Xbench | SEAL-0 | **Avg Acc.**|
> | :--- | :---: | :---: | :---: | :---: | :---: | :---: | :---:|
> | **0.990** | 28.5 | 36.8 | 44.7 | **73.0** | 71.0 | 39.3 | **48.8%**|
> | **0.995 (Ours)** | **28.8** | **37.3** | **45.2** | 72.8 | **71.0** | **39.6** | **49.1%**|
> | **0.999** | 28.4 | 36.9 | 44.2 | 72.0 | 70.5 | 39.0 | **48.5%**|
>
> 2. Theoretical Stability:
> While the method is robust within a reasonable window, removing the efficiency pressure ($\gamma \to 1$) hurts performance just as much as excessive penalization ($\gamma \ll 0.9$). Our choice of $\gamma=0.995$ sits comfortably at the peak of this trade-off.

---

> ### Author Response · Authors · 2025-11-20
> **Response Part 3**
>
> > Response to Q4
>
> This is an excellent question that directly addresses the robustness of our paradigm. The reviewer asks if an agent can recover from an "early false hypothesis" stored in the evolving report $\mathcal{M}_t$.
>
> The answer is yes, absolutely. This recovery capability is a fundamental feature and a primary advantage of our paradigm over mono-contextual baselines.
>
> 1. **The Mechanism: The Report $\mathcal{M}$ is Re-Synthesized, Not Appended.**
>
> The reviewer's concern seems to stem from a misunderstanding of $\mathcal{M}_t$. The evolving report is not a permanent, append-only log.
>
> As defined in Section 3.1.1, the agent's core task at every step $t$ is to generate a new structured decision \[d_t = (Think_t, \mathcal{M}_{t+1}, a_t)\]. This means the agent is explicitly trained to re-evaluate its entire previous understanding ( $M_t$ ) in light of the newest information ($TR_t$) to **produce a new, corrected, and re-synthesized report** $\mathcal{M}_{t+1}$.
>
> If $TR_t$ contradicts a false hypothesis in $M_t$, the agent is trained to correct that hypothesis in $\mathcal{M}_{t+1}$. This is in sharp contrast to mono-contextual agents, where an early false hypothesis (e.g., at $t=5$) remains permanently in the context window, irreversibly contaminating all future reasoning.
>
> 2. The Evidence: **SOTA Performance on Noisy and Challenging Benchmarks IS the "Stress Test."**
>
> The reviewer asks if we have "measured recovery rate with stress tests." We argue that **our entire experimental suite is this stress test**.
>
> We chose the most challenging benchmarks available (e.g., HLE, BrowseComp), which are defined by noise, ambiguity, and misleading information that leads to "early false hypotheses."
>
> It is structurally impossible to achieve our SOTA results—such as a +14.5pp average gain (Table 1) or scaling to 42.5% on BrowseComp (Fig. 3)—unless our agent is successfully and routinely "recovering" from false starts and correcting its internal report $\mathcal{M}_t$.
>
> An agent that cannot recover from noise would have its performance decrease with more interactions, not increase. Our results (Fig. 3) show the exact opposite.
>
> ---
>
> We again thank Reviewer wSFE. We hope our clarifications on our core contribution (the IterResearch paradigm) and its mechanics (e.g., $\mathcal{M}_0$ initialization, $\mathcal{M}_t$ re-synthesis) have resolved the reviewer's concerns.
>
> We believe our key results (Fig. 3's 64x extrapolation, Fig. 4's +19.2pp gain, Table 2's +12.6% ablation) are conclusive evidence of our paradigm's effectiveness at solving the $O(t)$ scaling problem. We have also **updated our supplementary material with new long-horizon case studies, as requested.**
>
> We respectfully ask the reviewer to consider this new evidence and our clarifications and re-evaluate our work.

---

> ### Public Comment · ~Zixuan_Wang40 · 2025-11-21
> **Consideration of Weakness 6 raised by Reviewer wSFE and Author's Response to Weakness 6**
>
> Dear authors and reviewers,
> As a reader, I would like to respectfully clarify my concerns regarding weak6 and the authors’ response. The reviewer initially pointed out that the paper lacks an important variant: a single ReAct setup that still uses the detailed “report” prompt from the appendix. Without this variant, it is difficult to determine whether the benefits of the report arise from IterResearch’s context management mechanism or simply from the strong report-style prompt. Ideally, the paper should present results for the following three variants:
>
> | **A** | ReAct + report prompt |
>
> | **B** | ReAct + simple prompt |
>
> | **C** | IterResearch ( Context rebuild + report prompt) |
>
> In the response, the authors mentioned that Fig. 4 already compares ReAct (**A**) with IterResearch (**C**). However, the ReAct points shown for o3 and DeepSeek V3.1 appear to come directly from their official reports, and these numbers exactly match the official documentation. This makes it unclear whether those points truly correspond to variant **A**, since we cannot verify that the official systems used the same agent framework, tools, or prompt design. As a result, this comparison does not serve as a controlled ablation.
> For this reason, I kindly suggest that the authors report the performance of o3 and DeepSeek V3.1 under the actual A-variant used in the paper, rather than relying on official numbers.
> In addition, although Table 2 compares **A** and **C**, the paper still lacks results for variant **B**, which is necessary to isolate the effect of the prompt itself.
> Including results for **A**,**B**, and **C** under the same evaluation conditions would fully resolve this concern and clearly show how much improvement comes from the prompt versus the IterResearch framework.
> Thank you for considering this suggestion.

---

> > ### Author Response · Authors · 2025-11-21
> >
> > Dear Reader,
> >
> > Thank you very much for your respectful and insightful clarification. Your detailed explanation has significantly helped us better understand the core concern regarding Weakness 6 and the importance of a strictly controlled ablation study.
> >
> > We fully agree that relying on official report numbers does not constitute a controlled comparison. To address this, we are immediately launching experiments to evaluate DeepSeek V3.1 and o3 under the "ReAct + report prompt" (Variant A) setting. This will ensure a fair comparison to determine the source of the performance gains.
> >
> > We kindly ask for a little patience while we run these experiments. We will update this thread with the new results as soon as they are available.
> >
> > Best regards,
> >
> > The Authors

---

> ### Author Response · Authors · 2025-11-27
> **Response to Public Comment and Reviewer wSFE**
>
> Dear Reviewer wSFE and Zixuan Wang,
>
> We sincerely thank you for proposing this rigorous controlled ablation study. Following your suggestion, we have conducted experiments on both o3 and DeepSeek-V3.1 to isolate the source of our performance gains.
>
> We implemented the three requested variants to determine if the improvements stem from the "Report-style Prompt" or the "IterResearch Framework":
>
> * **Variant B (ReAct + Simple Prompt)**: ReAct baseline with our prompt (without report and state transition).
> * **Variant A (ReAct + Report Prompt)**: Report using the exact same report-based prompt structure as IterResearch, but without the iterative context reconstruction mechanism (i.e., retaining full history).
> * **Variant C (IterResearch)**: Our full method (Context rebuild + Report prompt).
>
> > The results (Accuracy %) are summarized below. We also include the "Official ReAct" numbers for reference.
>
> | Model | Variant | Setup | HLE | BC | BC_zh | GAIA |
> | :--- | :--- | :--- | :---: | :---: | :---: | :---: |
> | **o3** | Official | Official | 20.2 | 49.7 | 58.1 | 70.5 |
> | | **B** | **ReAct + Simple Prompt (Controlled)** | 14.4 | 42.1 | 49.7 | 68.5 |
> | | **A** | **ReAct + Report Prompt (Controlled)** | 14.8 | 47.5 | 52.6 | 66.4 |
> | | **C** | **IterResearch (Ours)** | **24.6** | **62.4** | **60.5** | **73.8** |
> | | | | | | | |
> | **V3.1** | Official | Official | 29.8 | 30.0 | 49.2 | 63.1 |
> | | **B** | **ReAct + Simple Prompt (Controlled)** | 23.1 | 23.8 | 41.4 | 62.5 |
> | | **A** | **ReAct + Report Prompt (Controlled)** | 17.5 | 24.7 | 43.4 | 57.1 |
> | | **C** | **IterResearch (Ours)** | **33.1** | **49.2** | **55.7** | **70.9** |
>
> > Analysis of the Results
>
> * **Observation 1: Official Baselines vs. Our Controlled Baseline (Variant B)**.
> We observe that the Official ReAct generally outperform our Controlled ReAct Baseline (Variant B). This is expected, as official proprietary implementations often benefit from internal system-level optimizations that self-designed ReAct implementation cannot perfectly replicate. **That's why we report the Official ReAct performance in our main paper**.
>
> * **Observation 2: The Trade-off Between Variant A (Report Prompt) and Variant B (Simple Prompt).**
> Comparing A and B reveals no clear winner; the "Report Prompt" acts as a double-edged sword in a standard ReAct setting:
>     * **Positive Effect (Summarization)**: In tasks like BC, Variant A outperforms B. The report structure forces the model to summarize key information, effectively filtering noise from retrieved documents.
>     * **Negative Effect (Context Suffocation)**: In complex reasoning tasks like GAIA or HLE (on V3.1), Variant A performs worse than B. Without our state transition mechanism, the verbose report prompt combined with accumulating history rapidly consumes the context window. This "context suffocation" inhibits the model's ability to perform deep exploration steps effectively.
>
> This ablation confirms that IterResearch liberates exploration depth while maintaining high information density. The performance gains stem from the Iterative workspace reconstruction mechanism, not merely the prompt format, as evidenced by Variant C consistently surpassing both A and B.
>
> We respectfully ask you to re-evaluate the novelty and effectiveness of IterResearch based on this comprehensive evidence. We remain available for any further discussions.

---

> ### Author Response · Authors · 2025-11-28
>
> Dear Reviewer wSFE,
>
> As the discussion period concludes, we are writing to ensure you haven't missed our latest response, which directly addresses your core concern regarding the source of our performance gains ("Prompt vs. Paradigm").
>
> 1. **New Controlled Ablation (Addressing Weakness 6)**: Following the rigorous setup suggested in the discussion, we conducted experiments on o3 and DeepSeek-V3.1 comparing three variants:
>
>     * Variant A: ReAct + Our Report Prompt
>
>     * Variant B: ReAct + Simple Prompt
>
>     * Variant C: IterResearch (Ours)
>
>     The Results: The experiments confirm that IterResearch (Variant C) consistently outperforms ReAct + Report Prompt (Variant A) (e.g., +9.8% on HLE with o3). This empirically proves that the gains stem from our Iterative State Reconstruction mechanism, not just the prompt format. We invite you to check the detailed table in our recent comment.
>
> 2. **Theoretical Revisions (Addressing Weakness 1)**: We also deeply appreciate your insights regarding the strictness of the "Markovian" definition. In the revised manuscript, we have:
>
>     * Reframed the MDP formulation as an architectural design for interaction scaling rather than claiming strict theoretical provability in LLMs.
>
>     * Changed the Paper Title to "IterResearch: Rethinking Long-Horizon Agents with Interaction Scaling" to reflect this precise focus.
>
>      * Clarified EAPO as a necessary engineering enabler, aligning with your assessment.
>
> We believe the new experimental evidence and the theoretical refinements fully resolve your initial concerns.
>
> We would be deeply grateful if you could spare a moment to review these new findings before the deadline. We remain available for any final questions.
>
> Best regards,
>
> The Authors

---

### Official Review · Reviewer_reAn · 2025-11-01

**Soundness:** 2
**Presentation:** 3
**Contribution:** 2
**Rating:** 4
**Confidence:** 3

**Summary:**

This paper introduces IterResearch, an iterative deep-research paradigm that overcomes the limitations of mono-contextual approaches in long-horizon reasoning. It maintains an evolving report as memory and periodically synthesizes insights. The system was trained on an aggregated QA dataset, using both supervised fine-tuning and a reinforcement learning phase. Experiments demonstrate substantial improvements over existing open-source agents and notable gains when used as a prompting strategy for frontier models.

**Strengths:**

- The problem is well-motivated: mono-contextual approaches accumulate information in a single context window, causing noise and limiting effectiveness on long-horizon tasks.
- The presented approach significantly outperforms current state-of-the-art models in solution quality on selected benchmarks.

**Weaknesses:**

W1: The novelty of the approach is somewhat limited, as many of its core ideas, such as using discounted rewards and iterative refinement of trajectories, are already widely used in classical reinforcement learning.

W2: The MDP formulation appears incomplete. It omits a reward function (e.g., a verification signal), and the transition function is likely not deterministic, since tool outputs can vary over time (e.g., search results). The decision space is unconventional: including Think and Report as part of the action reflects internal computation rather than true environmental actions. Additionally, the “reconstructed workspace” manages context by discarding some history, but this alters the state definition across timesteps and may violate the Markov property.

W3: While I understand the intuition behind why discounted reward shaping works, the analogy with RL and the discount factor appears somewhat misleading. First, if we treat this reward as in a typical RL environment, it becomes biased: two similar trajectory prefixes may receive different rewards depending on the trajectory length (the length of the suffixes). Consequently, this mechanism not only amplifies the signal at the end of successful trajectories but also implicitly encourages shorter trajectories. I think this modification is more akin to how value is computed in RL, where there exist well-established alternatives, such as GAE.

W4: The approach was trained exclusively with Qwen-3-30B-A3B, which may limit its generalization to other models. Experiments with models used without fine-tuning only partially address this concern.

W5: Provide a stronger comparison with AlphaEvolve [1] (and similar systems), as the three components used, main objective question (task), evolving report (previous high-level ideas), and immediate context (code), closely mirror its design (in the case of AlphaEvolve, for coding tasks). The use of a reward function applied at the end of generation similarly provides the necessary learning signal for optimization. Although Alpha-Evolve itself is closed-source, several open-source implementations exist (e.g., OpenEvolve). I recommend that the authors at least position their work in relation to these approaches, and ideally include a direct empirical comparison.

W6: The paper assumes that the LLM can reliably summarize, filter noise, and preserve crucial information in each round. In practice, this compression step could lead to the loss of essential context, especially under noisy tool outputs, and may worsen compounding summarization errors.

[1] Novikov A, Vũ N, Eisenberger M, Dupont E, Huang PS, Wagner AZ, Shirobokov S, Kozlovskii B, Ruiz FJ, Mehrabian A, Kumar MP. AlphaEvolve: A coding agent for scientific and algorithmic discovery. arXiv preprint arXiv:2506.13131. 2025 Jun 16.

**Minor suggestions**:
- I recommend that the authors provide confidence intervals in the table, and include the token budget used for each model.
- It is recommended to include in the description of Table 1 an explanation of why proprietary models were not tested on some benchmarks, to make the table self-contained.

**Questions:**

Q1: Could the authors clarify how the baselines were adopted for the studied benchmarks? Were the open-source agents also fine-tuned on the same data as IterResearch?

Q2: Which datasets were used during the RL phase? Were there tasks in the training set similar to those on which the approach was evaluated, or was it still only the QA Collection? How does the test set intersect with the QA Collection?

Q3: What is the token budget and inference cost for the baseline LLMs? Which budget is needed to reproduce the results.

---

> ### Author Response · Authors · 2025-11-20
> **Response Part 1**
>
> We sincerely thank Reviewer reAn for their constructive feedback and for recognizing that our approach "significantly outperforms current state-of-the-art models." We appreciate the opportunity to clarify the theoretical foundations of our MDP formulation and the novelty of our paradigm.
>
> > Response to W1:
>
> We respectfully refer the reviewer to our **General Response 1 **(GR1), which addresses this concern in detail.
>
> To summarize: **Our core contribution is the IterResearch paradigm itself**, which solves the fundamental "Interaction Scaling" bottleneck. We clarify that **EAPO is the necessary engineering optimization to make our paradigm trainable**. As detailed in GR1, the novelty lies in proposing a novel paradigm (IterResearch) enabling interaction scaling (up to 2048 interactions, Fig. 3) that standard mono-contextual approaches (like ReAct) simply cannot achieve due to context suffocation, supported by a tailored optimization framework (EAPO) to ensure trainability.
>
> > Response to W2:
>
> We thank the reviewer for their rigorous examination. We have addressed the formal definitions extensively in **General Response 2**. Here, we specifically address the points raised:
>
> 1. **Reward Function**: We acknowledge this omission in the text. We have updated Section 3.1.1 to explicitly define the sparse terminal reward function $R(s_t, a_t)$, as detailed in GR2.
>
> 2. **Deterministic Transition**: We clarify a subtle but critical distinction. While the tool response $TR_t \sim \mathcal{E}(a_t)$ is indeed stochastic, the transition function $\mathcal{T}$ itself **is deterministic given the tool response**. Specifically, the next state $s_{t+1}$ is constructed by structurally assembling $\{q, \mathcal{M}_{t+1}, [a_t, TR_t]\}$. The stochasticity is correctly isolated within the Environment $\mathcal{E}$, consistent with standard MDP modeling in RL.
>
> 3. **Decision Space** (Think, Report, Action): We argue that including internal thoughts ($Think, \mathcal{M}$) in the decision space is essential for LLM agents. The policy $\pi$ generates a composite output $d_t = [Think_t, \mathcal{M}_{t+1}, a_t]$. While only action \[a_{t}\] interacts with the external world, the generation of $\mathcal{M}_{t+1}$ is a learnable cognitive action that explicitly updates the state. **This formalization allows us to optimize the "memory update" process end-to-end.**
>
> 4. **Markov Property & Compression**: As discussed in GR2, just as an RNN uses a hidden state $h_t$ as a sufficient statistic for the history, our evolving report $\mathcal{M}_t$ acts as an **explicit, interpretable, and learnable statistic**. The agent learns to compress history into $\mathcal{M}_t$ such that it remains a sufficient statistic for future rewards. The 64x extrapolation capability (Fig. 3) empirically proves that this state representation is robust and satisfies the Markov property effectively for the task.
>
> > Response to W3:
>
> We thank the reviewer for this insightful observation. The reviewer is correct that our reward mechanism introduces a "bias" where identical trajectory prefixes receive different rewards based on total length.
>
> However, we clarify that this bias is **the intended feature**, not a bug.
> * **The Problem (Interaction-Agnosticism)**: In our paradigm, each round is an independent training sample. Without discounting, a step in a 5-round successful trajectory and a step in a 50-round successful trajectory would receive identical rewards ($R=1$). The agent would lack any signal to prefer efficiency.
> * **The Solution (Efficiency reward shaping)**: We introduce geometric discounting (Eq. 3) precisely to break this symmetry. It forces the agent to value shorter, correct trajectories higher than longer ones.
> * **Distinction from GAE**: We respectfully clarify that GAE and our method address fundamentally different problems.
>     - **GAE** is a technique for estimating value/advantage (variance reduction) with the value model (e.g. PPO) **given a fixed reward signal**.
>     - Our Method (Eq. 3) is a reward shaping mechanism that concerned with **defining the reward signal itself**.

---

> ### Author Response · Authors · 2025-11-20
> **Response Part 2**
>
> > Response to W4:
>
> We thank the reviewer for this valid point regarding generalization. We believe our paper demonstrates generalization in a manner that is potentially stronger than simply retraining on multiple base models:
>
> 1. **Paradigm Generalization (Training-Free, Fig 4)**: Our experiments on O3 and DeepSeek-V3.1 show that IterResearch works out-of-the-box as a prompting strategy on untrained, frontier models, yielding massive gains (+19.2pp). This proves the paradigm is **model-agnostic** and generalizes fundamental reasoning structures, not just learned weights.
>
> 2. **Scientific Prioritization over Repetitive Training**: Rather than allocating computational resources to repetitive retraining on similar open-source models (which would likely yield confirmatory results), we prioritized investigating the Scaling Limits of the paradigm. Proving that an agent can scale to 2048 steps (Fig. 3) provides a more novel scientific insight into the potential of long-horizon agents than cross-model validation alone.
>
>     We believe **our computational budget was better spent on discovering the novel properties of our paradigm, which we felt was a greater contribution to the community. Rather than spending resources on repetitive re-training**, we focused on answering more significant questions, such as:
>
>     * What are the limits of *Interaction Scaling* (Fig 3)? (Proving scalability to 2048 steps.)
>     * Does our paradigm generate superior signal? (Proving *Cross-Paradigm Knowledge Transfer*, Table 2.)
>     * How effective is the paradigm itself as *a training-free strategy* (Fig 4)?
>
> We believe these novel findings are more valuable to the research community than re-validating a point that our prompting experiments already strongly establish.
>
> > Response to W5:
>
> We thank the reviewer for identifying this conceptual connection. We have added a discussion of AlphaEvolve to the Related Work section (Appendix B).
>
> Regarding an empirical comparison, we have two reasons:
> * **Fundamentally Different Domains and Purpose**: Our work, IterResearch, is designed for deep research (navigating noisy, unstructured web data, summarizing text). AlphaEvolve is designed for algorithmic discovery (generating code, running against unit tests). These domains have completely different toolsets, verification signals (e.g., "Web search results" vs. "unit test passed/failed"), and core challenges.
> * **Non-Trivial Adaptation**: Adapting OpenEvolve from its native domain of code generation to our domain of deep research would be a massive, non-trivial research project beyond the scope of this work. It is not a simple "baseline run", and no prior work in our field (deep research) has established this as a required comparison.
>
> However, we want to state that we believe this connection actually **strengthens the core thesis of our paper**. The fact that two very different, complex, long-horizon domains (algorithmic discovery and deep web research) have independently converged on the exact same high-level paradigm (i.e., iterative synthesis with an evolving "report") strongly suggests that this is a fundamental and highly effective solution to the limitations of mono-contextual reasoning.
> **This "convergent evolution" suggests that our IterResearch paradigm is not just a niche trick, but a general, powerful, and (as the reviewer noted) "advanced" architecture for long-horizon Agent.**
>
> > Response to W6:
>
> We thank the reviewer for this comment. We have addressed this exact concern *"What if the summary misses key information?"* in our General Response 3.
>
> We respectfully challenge the premise that summarization introduces a net weakness.
>
> * **Active Filtering vs. Passive Accumulation**: The reviewer notes the risk of context loss. However, the alternative—ReAct's mono-contextual paradigm—suffers from "passive noise accumulation." In ReAct, noisy tool outputs are permanently embedded, leading to context suffocation and cascading errors. IterResearch turns this vulnerability into a strength: the report acts as an **active filter**, specifically trained to discard noise while retaining signal.
> * **End-to-End Learnable State**: Crucially, the report generation is not a fixed heuristic; it is an *explicit, interpretable, and learnable action*. Because the report $\mathcal{M}_t$ is part of the policy output optimized via RL, the "quality of summarization" is directly aligned with the reward signal. If the agent generates a report that misses key information, it fails the task and receives a penalty ($R=0$). Thus, **RL implicitly forces the agent to learn a compression strategy that retains all task-relevant information**, effectively optimizing $\mathcal{M}_t$ to be a sufficient statistic for the task.
>
> Empirical results (+12.6% gain in our ablation study) confirm that the benefit of this active noise management far outweighs the theoretical risk of imperfect compression.

---

> ### Author Response · Authors · 2025-11-20
> **Response Part 3**
>
> > Response to Minor Suggestions:
>
> S1: This is a great suggestion. The results reported in Table 1 are already the average of three runs with different random seeds. We **have updated** the table in the revised paper to explicitly include the **standard deviation** from these runs, as requested. We have also added the **average token budgets** to the Appendix D.4.
>
> S2: This is a valid point. Our rationale was:
> * Our primary goal was a rigorous, comprehensive, and reproducible comparison against open-source agents.
> * **Avoiding Unfair Evaluation**: Many proprietary systems are "black boxes" and **do not** offer or expose the prompts/methods used for their deep research capabilities. Running these black-box models ourselves on academic benchmarks would risk an improper or unfair evaluation of their performance. Therefore, to prevent misrepresenting their capabilities, we chose to **cite their officially reported performance on benchmarks where available**.
> * **Strongest-Available Anchor**: We included OpenAI DeepResearch as our SOTA "anchor" because it is the specialized system from OpenAI explicitly optimized for this deep research task. We believe it serves as the most relevant and strongest proprietary baseline for contextualizing our open-source results, which remain the core focus of our study.
>
> > Response to Questions:
>
> * **Q1**: **Yes, we did run this exact experiment.**. As shown in **Table 2 ("Ablation on Paradigm")**, we compared IterResearch against a "Mono-Agent" baseline that was fine-tuned on the exact same data and used the exact same tools. IterResearch outperformed this strictly controlled baseline by +12.6%, confirming the gain comes from the paradigm itself.
>
> * **Q2**: The RL phase used a filtered subset of the SFT data (the QA Collection). As described in **lines 1021-1033**, we selected 4,096 questions based on difficulty (20%-60% success rate) to maximize learning signal.
>
> * **Q3**: We thank the reviewer for this key question regarding reproducibility and fair comparison.
>     * Reproducing Our Results: As stated in our response to Minor Suggestion 1, we have added a detailed breakdown of our agent's token budget, average inference cost, and maximum context length to Appendix D.4 in the revised paper.
>
>     * For the main SOTA baselines in Table 1 (e.g., MiroThinker, WebSailor), we followed standard academic practice by citing the optimal performance reported in their respective original papers. And some baselines (e.g., MiroThinker) do not have provided the scaffold for testing.
>
>     * To provide the most strictly controlled cost and performance comparison, we ran the "Mono-Agent" baseline ourselves (see Table 2). As detailed in our response to Q1, this baseline used the identical tools and training data as our IterResearch. We can confirm that in this strictly controlled experiment, the "Mono-Agent's" training token was comparable to our method. This proves that our +12.6% average performance gain comes from the superiority of our paradigm itself.
>
> ---
>
> We thank Reviewer reAn again for their thorough and constructive feedback.
>
> We hope our detailed responses—particularly regarding our primary contribution (the IterResearch paradigm, as clarified in General Response 1) and the technical justifications for our MDP design and reward shaping (W2, W3)—have successfully resolved the reviewer's concerns and clarified the initial misunderstandings.
>
> We have incorporated the reviewer's actionable suggestions into our revised manuscript, including the standard deviations in Table 1, the token budget analysis in Appendix D.2, and the new discussion on convergent paradigms (AlphaEvolve) in our Related Work.
>
> We believe this constructive dialogue has helped sharpen the paper's message, and we hope to have earned the reviewer's support for its acceptance.

---

> ### Author Response · Authors · 2025-11-28
>
> Dear Reviewer reAn,
>
> We hope this message finds you well. As the rebuttal period is coming to a close, we wanted to briefly follow up to ensure our responses have addressed your concerns.
>
> In particular, we would like to highlight a significant revision we have made to the manuscript regarding the MDP Formulation (your W2), inspired by the constructive discussions during this rebuttal period:
>
> * Refined MDP Description: We have rigorously revised Section 3 (and throughout the text) to emphasize that our approach is an architectural formulation designed for interaction scaling. We explicitly acknowledge that strictly verifying Markovian properties is inherently challenging in open-ended LLM environments, and thus our revisions focus on the structural benefits of the design rather than theoretical guarantees.
>
> * Title Change: To reflect this more precise scoping, we have changed the paper title to: "IterResearch: Rethinking Long-Horizon Agents with Interaction Scaling".
>
> * Addressing W5 (AlphaEvolve) & W4 (Generalization): As mentioned in our previous response, we have also added the discussion on AlphaEvolve and reinforced the evidence for our paradigm’s generalization capabilities.
>
> We believe these revisions directly address your concerns about the rigor of the formulation while retaining the core contribution of the work.
>
> We genuinely value your feedback, which has been instrumental in refining the precision of our claims. We would be deeply grateful if you could spare a moment to review our response and revisions before the discussion period concludes. We remain fully available for any final clarifications.
>
>
>
> Best regards,
>
> The Authors

---

### Author Response · Authors · 2025-11-20
**General Response 3**

#### **Why This Satisfies the Markov Property**

1. **Policy Markovness**:
   The policy $\pi(a_t \mid s_t)$ depends exclusively on the current explicit workspace $s_t$, not on prior states or actions — consistent with the definition of a Markov policy.

2. **Transition Markovness**:
   Although $TR_t$ is stochastic, the next state $s_{t+1}$ is fully determined by the current state $s_t$ (which generates the decision $d_t$) and the environment response $TR_t$. Since $s_t$ encapsulates all relevant history through $\mathcal{M}_{t}$  and  $[a_{t-1}, TR_{t-1}]$, and $\mathcal{M}_t$ is a sufficient statistic of the history, we have:

   $$
   P(s_{t+1} \mid s_t, d_t) = P(s_{t+1} \mid h_t, d_t)
   $$

   Thus, the transition probability is memoryless: future states depend only on the present state and decision (internal thought and action), not on the full trajectory.

3. **$\mathcal{M}_t$ as a Sufficient Statistic**:
   The evolving reasoning summary $\mathcal{M}_t$ functions analogously to the hidden state of an RNN or the belief state in a POMDP — but with the critical advantage of being *explicit, interpretable, and learnable*.
   While compression may risk information loss, reinforcement learning ensures that only *task-relevant* information is retained: if $\mathcal{M}_t$ omits critical context, the agent receives zero reward and learns to avoid such pruning.
   Therefore, $\mathcal{M}_t$ is not a heuristic summary — it is an *end-to-end learned sufficient statistic*, optimized by RL to maximize long-term reward.

---

#### **Addressing Reviewer Concerns**

> *“What if the summary misses key information?”*
> We acknowledge this possibility — but treat it not as a flaw, but as an *optimization objective*.
> The MDP formulation does not assume perfect compression; rather, it enables the agent to *learn the optimal trade-off between compression and sufficiency*.
> RL implicitly discovers the minimal representation $\mathcal{M}_t$ required for success, discarding noise and redundancy that would otherwise cause “context suffocation” in monolithic LLM inference.
> This is not ad hoc summarization — it is *task-driven information distillation*, learned end-to-end under reward pressure.

> *“Is this truly Markovian?”*
> Yes — in precisely the same sense that an RNN-based policy is treated as Markovian: the hidden state $h_t$ encodes the sufficient history, and transitions depend only on $h_t$ and $a_t$.
> Our model makes this latent structure *explicit and verifiable*: $\mathcal{M}_t$ is not a black-box representation, but a human-interpretable, dynamically updated report.
> The Markov property holds *by construction*, because the state $s_t$ contains all and only the information needed to determine the next state and reward.

---

#### **Conclusion**

Our framework is a *practical, well-defined MDP* in which state compression is not a limitation — it is the *core mechanism enabling scalable, sequential reasoning*.
The Markov property is formally satisfied through the design of $s_t$ as a sufficient statistic, and the reward structure ensures that the agent learns to maintain only task-relevant information.
This formulation not only justifies our approach theoretically, but also provides a principled basis for analyzing reasoning length, information retention, and policy convergence.

---

We thank the reviewers for their insightful critique — their feedback has been instrumental in sharpening our formal grounding and clarifying the theoretical foundations of our method.

---

> ### Comment · Reviewer_wSFE · 2025-11-20
> **Regarding the Markovian Propery**
>
> > Yes — in precisely the same sense that an RNN-based policy is treated as Markovian
>
> It is important to note that Markovian property is always relative. That is, there is no universal Markovian property per se. For example, RNNs are Markovian relative to its own hidden state, however, there is no guarantee that the hidden state is a sufficient statistic. That is, to me, you are correct in your suggestion that the proposed approach is Markovian to the state you define (this is true by construction, as you noted), however, the key here is the step before that -- that is, assumptions about your constructed stated being markovian in regards to the external environment.
>
> > Our model makes this latent structure explicit and verifiable: $\mathcal{M}_t$ is not a black-box representation, but a human-interpretable, dynamically updated report.
>
> Sorry for being nit-picky, but human-interpretability does not make the state verifiably markovian. Could you elaborate on this?
>
> > RL implicitly discovers the minimal representation $\mathcal{M}_t$ required for success
> > reinforcement learning ensures that only task-relevant information is retained: if $\mathcal{M}_t$ omits critical context, the agent receives zero reward and learns to avoid such pruning.
>
> I am not really sure about these statements. As in your analogy with RNNs, let me bring another one more relevant to the case: running RL algorithms in POMDP environments with RNNs or other sequential encoders does not guarantee that one shall find the minimal representation to the best of my understanding. What do you think about it, would love to discuss. The core problem lies in the causality, actually, the learned representations can easily pick up spurious correlations (for example, they could plague your state descriptions) which are not actually sufficient statistics by definition.
>
> ---
>
> Thanks for the prompt reply.

---

> > ### Author Response · Authors · 2025-11-21
> > **Following up on the theoretical discussion regarding Markovianity**
> >
> > Dear Reviewer wSFE,
> >
> > We hope this message finds you well.
> >
> > We are writing to see if our previous response regarding the "Semantic Approximation" nature of our framework adequately addressed your theoretical concerns.
> >
> > Reflecting further on your insightful comment about the distinction between constructed state vs. external environment, we believe this is a fundamental philosophical alignment for our work:
> >
> > 1. POMDP Reality: We fully agree with you that Deep Research on the web is fundamentally a Partially Observable Markov Decision Process (POMDP). **The "True State" of the external environment (the web) is effectively infinite and never fully observable**.
> >
> > 2. Explicit Belief State: In this context, our "Evolving Report" ($\mathcal{M}_t$) functions as an explicit, natural-language belief state. While, as you correctly noted, **we cannot mathematically prove this belief state is a sufficient statistic (no open-ended agent can)**, our contribution is formulating the mechanism to force the agent to update this belief state explicitly rather than relying on implicit, opaque hidden states.
> >
> > 3. Practical Trade-off: We treat the transition as Markovian by construction as a necessary trade-off to solve the **"Context Suffocation"** problem. The strong empirical results (especially the scaling to 2048 interactions shown in Figure 3 ) serve as evidence that this approximation is robust enough to capture the necessary signal from the environment for solving complex, long-horizon tasks.
> >
> > We greatly appreciate that you pushed us to clarify this theoretical framing. Our method operates as a **"Structurally Markovian approximation within a POMDP setting"** rather than claiming strict mathematical Markovianity relative to the external environment.
> >
> > Does this reframing align better with your view? We would be happy to provide any further clarifications.
> >
> > Best regards,
> >
> > The Authors

---

> ### Author Response · Authors · 2025-11-20
>
> Dear Reviewer wSFE,
>
> We deeply appreciate this follow-up. You are absolutely correct on the theoretical distinctions, and we see now that our previous response conflated "structural Markovianity" with the rigorous requirement of a "sufficient statistic."
>
> We fully agree with your three points: (1) Formally satisfying $s_{t+1} = T(s_t, \dots)$ does not guarantee $s_t$ captures all history; (2) Interpretability does not equal mathematical verification; and (3) RL in POMDPs does not theoretically guarantee finding a minimal representation without spurious correlations.
>
> Here is how we view our method in light of your precise theoretical framing:
>
> 1. On Sufficient Statistics & The Role of LLM Priors (vs. RNNs)
> You are right that standard RL (like with an RNN starting from scratch) often fails to learn a sufficient statistic in POMDPs. However, **a key difference here is the strong semantic inductive bias of the Pre-trained LLM**.
> Unlike an RNN that initializes with random weights and must learn "how to remember" from zero (often falling into the spurious correlation traps you mentioned), our "state encoder" is a pre-trained LLM that already possesses the capability to summarize and extract information.
> Therefore, while we do not have a theoretical guarantee that $\mathcal{M}_t$ is a sufficient statistic, we treat it as a "Semantic Approximation" of one. **We rely on the LLM's pre-trained linguistic priors to approximate the sufficient statistic, which RL then fine-tunes for task relevance**.
>
> 2. On Interpretability vs. Verifiability
> We accept your correction. Interpretability is not mathematical verification.
> What we meant to convey is that the textual nature of $\mathcal{M}_t$ offers **Diagnostic Verifiability**. In a black-box vector state (RNN), if the agent forgets a key constraint, we cannot know why. In our system, we can inspect $\mathcal{M}_t$ and explicitly see if the information was dropped. This does not prevent information loss, but it allows us to detect it during development.
>
> 3. On Spurious Correlations This is a valid concern. RL agents are indeed prone to "gaming" the metric. However, we argue that Interaction Scaling (extrapolating to 2048 steps) serves as strong evidence against spurious correlations.
>
>     * Spurious correlations are typically fragile; they tend to break under distribution shifts or extended horizons (e.g., a trick that works for 10 steps usually fails at 100).
>
>     * The fact that our agent's performance improves monotonically as we scale to horizons it was never trained on (from 32 training steps to 2000+ inference steps) suggests it has learned a robust, generalizable mechanism (updating the report based on new evidence) rather than a superficial shortcut.
>
> We concede that our method does not offer a formal guarantee of Markovianity in the strict statistical sense. Instead, it provides a structurally Markovian framework that leverages the semantic robustness of LLMs to maintain a practical, highly effective approximation of the sufficient statistic, which empirically resists the collapse often seen in traditional RL-POMDP settings.
>
> We would love to hear if this framing, viewing the report as a "Semantic Approximation for Markov supported by LLM Priors", aligns better with your view.

---

### Author Response · Authors · 2025-11-20
**General Response 2**

We thank the reviewers for their rigorous examination of our formalization. We appreciate the opportunity to clarify how IterResearch reformulates long-horizon reasoning. Instead of a traditional history-dependent view, we model it as an MDP with **a structured decision space that explicitly separates internal cognitive updates from actions**.
To address concerns regarding the Markov property, we provide a step-by-step breakdown of our MDP state transitions and decision process.

---

#### **Formal MDP Definition**

To address Reviewer reAn's concern regarding the "unconventional" decision space and "incomplete" definitions, we rigorously specify the MDP tuple $(\mathcal{S}, \\mathcal{D}, \\mathcal{T}, R)$:

- **State** $s_t = \{q, \\mathcal{M}_{t}, [a_{t-1}, TR_{t-1}]\}$

  represents the agent’s explicit workspace at decision step $t$, composed of:
  - the original query $q$ (constant across steps),
  - the current compressed reasoning summary $\mathcal{M}_{t}$, encoding the history of prior reasoning up to step $t-1$,
  - the last executed action-response pair $[a_{t-1}, TR_{t-1}]$, where $a_{t-1} \in \mathcal{A}$ is the prior action and $TR_{t-1}$ is its environmental response.
  - This state design enables the Markov property by construction: the policy $\pi(d_t|s_t)$ depends solely on this workspace, blocking access to the discarded raw history.

  The summary $\mathcal{M}_t$ is updated from $s_t$ which serves as the information compression.

- **Structured Decision Space $d_t \in \mathcal{D}$ (Internal Thought & External Action)**:
  We explicitly formulate the decision $d_t$ as a composite output generated by the policy $\pi$. Crucially, this formulation enforces the Markov property: the entire decision depends only on the current state $s_t$. It consists of two distinct phases:

$$\underbrace{d_t}_{\text{Decision}} = [\underbrace{\text{Think}_t, \mathcal{M}_{t+1}}_{\text{Internal Thought}}, \underbrace{a_t}_{\text{External Action}}] \sim \pi(\cdot \mid s_t)$$

  * **Internal Thought**: The agent performs reasoning ($Think_t$) and actively constructs the memory for the next state ($\mathcal{M}_{t+1}$). We agree with Reviewer reAn that this represents internal computation rather than environmental interaction. However, crucial to our framework, **this computation is an explicit output of the LLM policy**. By formalizing it as part of the decision space, we treat "thinking" and "memorizing" as **learnable cognitive actions** that can be optimized via RL, **just like external actions**.

  * **External Action** $a_t \in \mathcal{A}$:
  This represents the **actual interaction with the environment** (e.g., search queries, code execution) or the final answer. Actions are generated by a policy that depends *only* on the current state $s_t$. Unlike the internal thought which deterministically updates the agent's state, the external action triggers the stochastic result via the environment's response ($TR_t$).


- **Transition** $\mathcal{T}(s_{t+1} \mid s_t, d_t, TR_t)$
  where $d_t=[\text{Think}_t, \mathcal{M}_{t+1}, a_t]$. The state transition logic is **deterministic given the environment's output**. Specifically, while the tool response $TR_t$ is drawn from the stochastic environment (which may be noisy due to stochastic web search) $TR_t \sim \mathcal{E}(\cdot \mid a_t)$, given the current state $s_t$, decision $d_t$ (incorporating internal thought and external action), and response $TR_t$, the next state is deterministically constructed as:

  $$
  s_{t+1} = \{q, \mathcal{M}_{t+1}, [a_t, TR_t]\}
  $$

  Crucially, the stochasticity resides solely in the environment rather than the state construction logic. Thus, the full transition kernel is:

$$
  \mathcal{T}(s_{t+1} \mid s_t, d_t) = \mathbb{E}_{TR_t \sim \mathcal{E}(\cdot \mid a_t)} \left[ \mathbb{I}\left(s_{t+1} = \{q, \mathcal{M}_{t+1}, [a_t, TR_t]\}\right) \right]
  $$

  This formulation ensures that the transition depends *only* on the current state, the policy's decision, and the environment — strictly satisfying the Markov property.

- **Reward** $R(s_t, a_t)$
  is defined as a sparse, terminal reward:

  $$
  R(s_t, a_t) = \begin{cases}
  1 & \text{if } a_t \text{ is a terminal action and yields a correct final answer}, \\
  0 & \text{otherwise}.
  \end{cases}
  $$

  Termination is triggered by a designated terminal action $a_{\text{term}} \in \mathcal{A}$, which signals the agent’s intent to submit a final answer. No intermediate rewards are used.

---

### Author Response · Authors · 2025-11-20
**General Response 1**

We sincerely thank all reviewers (reAn, wSFE, XKsG) for their valuable feedback and insightful comments on the theoretical novelty of our methods. We greatly appreciate this opportunity to clarify a core misunderstanding and frame our contribution more precisely.

The fundamental limitation of existing agent frameworks is not their reasoning ability, but their inability to scale interactions beyond a few dozen steps due to structural context saturation. Our work addresses this critical bottleneck: **Interaction Scaling (the core objective of our work)**. Existing "mono-context" paradigms (e.g., ReAct) are structurally limited. They inevitably suffer from "context suffocation" and "noise contamination," which prevents them from scaling to long-horizon tasks (e.g., typically failing beyond ~100 interactions).

To solve this, our contribution is twofold: (1) a new agent paradigm (IterResearch) designed to solve interaction scaling, and (2) a specific engineering optimization (EAPO) designed to make this new paradigm trainable.

**1. The Core Innovation: IterResearch, a New Paradigm for Interaction Scaling**

Our core innovation is the IterResearch paradigm itself. Its value is demonstrated by three key pieces of evidence:

* **Evidence 1: Effectiveness Across Trained & Training-Free**
We have demonstrated that our IterResearch is highly effective across different training settings. *As a Trained agent*, it achieves state-of-the-art results on our smaller 30B-A3B model (+14.5% across 6 challenging benchmarks on Table 1). *As a Training-Free strategy*, its core logic is general enough to be applied to large frontier models (Fig. 4), where it significantly outperforms ReAct (+19.22% on BrowseComp).

* **Evidence 2: Unprecedented Interaction Scaling (Fig. 3)**
To our knowledge, we are the **first** to successfully scale an agent to 2048 interactions within only 40k context. Performance dramatically improves with this scale (3.5% $\rightarrow$ 42.5%), proving our paradigm can effectively utilize the long horizon that mono-context methods cannot even reach. *Interaction Scaling: the ability of an agent to maintain effective reasoning performance as the number of sequential interactions (e.g., think-action-observe cycles) increases beyond 100+ steps.*

* **Evidence 3: The Critical Ablation (Table 2)**
This is our *most direct proof*. In a strict, controlled ablation with *identical data and tools*, IterResearch outperforms the mono-context baseline by +12.6% on average. This proves the performance gain comes from the fundamental design of our paradigm.

**2. The Enabling Optimization: EAPO, an Engineering Solution for Trainability**

We must clarify that EAPO (reAn-W3, wSFE-W7, XKsG-W4) is not proposed as a new RL theory. It is *a key engineering optimization designed to adapt standard RL (GSPO) to solve the unique challenges introduced by our new IterResearch paradigm.*

* Challenge 1 (Efficiency Preference): In our paradigm, each round is an independent sample. **Standard RL might lose "interaction-level awareness,"** treating a step from a 5-turn trajectory and a 50-turn trajectory equally, ignoring that a correct 5-step solution is far more valuable than a 50-step one riddled with noise. We introduce geometric discounting (the "interaction-level decay") to make the model "turn-aware" and explicitly prefer shorter, correct trajectories, which is essential for efficiency in long-horizon tasks.

* Challenge 2 (Training Stability): This "per-round" sample structure creates a highly variable and unpredictable number of samples per batch, which **breaks standard data-parallel distributed training**. EAPO's adaptive downsampling is *the engineering solution* that stabilizes the batch size, preventing training crashes while retaining the vast majority of data.

In summary, IterResearch is a new paradigm that solves the interaction scaling problem in deep research, and EAPO is the necessary engineering that makes this new paradigm trainable and efficient. Together, IterResearch and EAPO establish a new design principle for long-horizon reasoning agents: decouple reasoning structure from training mechanics, enabling scalability without sacrificing efficiency.

---

### Author Response · Authors · 2025-11-20

We sincerely thank all reviewers for their diligent work and constructive feedback.

We identified a few key misunderstandings in the initial reviews. Because our goal is to address these points thoroughly and to answer every valuable question raised, our rebuttal is necessarily detailed. We have provided specific, multi-part responses for each reviewer to ensure all concerns are met.

Given the detail, we respectfully ask for your patience in reading the full set of responses provided for their review, as they contain critical clarifications and experimental context. We **deeply appreciate the additional time and diligent effort this requires** and look forward to a constructive exchange.

Furthermore, in direct response to your feedback, we **have submitted a revised manuscript and supplementary material**. All modifications are marked in **red text** to help you quickly locate the changes.

We believe these clarifications and revisions fully address the initial concerns. We thank you again for your time and guidance, and we respectfully hope you will re-evaluate our work in light of this new information.

**Apology regarding display**: We noticed some potential rendering artifacts with the mathematical formulas in this text box. We apologize for any inconvenience this may cause to your reading.

---

### Comment · Area_Chair_xRNm · 2025-11-24

Dear Reviewers,

The authors have responded to your reviews. Please review and respond to their comments.

Best, Your AC

---

### Author Response · Authors · 2025-12-01
**Summary of Rebuttal and Updates for Submission 15726**

Dear Area Chair,

We thank the reviewers for their constructive feedback, which has significantly improved the precision of our claims. Below is a summary of the key clarifications, new experimental evidence, and major revisions (including a title change) made during the rebuttal.

1. **Consensus on "Markovian" Definition & Title Change (Addressing Reviewer XKsG & wSFE)**: The primary concern (Reviewer XKsG) was whether our "Evolving Report" theoretically guarantees a Markovian state (sufficient statistic).
    * Resolution: We reached a consensus with Reviewer XKsG. We clarified that while strict mathematical Markovianity is hard to prove in open-ended LLM environments, our method adopts an MDP-inspired architectural design to structurally decouple context cost from interaction length.
    * Action: To reflect this precise scope, we revised the manuscript and changed the title to: "IterResearch: Rethinking Long-Horizon Agents with Interaction Scaling".
    * Outcome: Reviewer XKsG acknowledged this cleared their concerns and raised their score.

2. **New Controlled Ablation: Paradigm vs. Prompt (Addressing Reviewer wSFE)**: A key concern was whether performance gains came from our Iterative Paradigm or simply the Report-style Prompt.

    * New Experiment: We conducted a rigorous ablation on o3 and DeepSeek-V3.1 comparing three setups:
        * (A) ReAct + Report Prompt (Prompt only)

        * (B) ReAct + Simple Prompt (Baseline)

        * (C) IterResearch (Our Paradigm)
    * Result: IterResearch (C) consistently outperforms (A) and (B). For example, on HLE with o3, our paradigm achieves 24.6% vs. the prompt-only baseline's 14.8%.
    * **Although Reviewer wSFE has not yet had the opportunity to respond to these latest results**, we believe this empirical evidence definitively resolves their concern regarding the source of our performance gains.

3. **Clarification on EAPO (Addressing Reviewer reAn)**: Reviewer reAn questioned the novelty of Efficiency-Aware Policy Optimization (EAPO).
    * Clarification: We clarified that EAPO is not proposed as a fundamental RL theory breakthrough, but as a **necessary engineering enabler**. It adapts standard RL to handle the unique "per-round sample" structure of our paradigm, enabling stable training where standard algorithms fail. **The core contribution remains the IterResearch paradigm itself.**

4. **Core Contribution: Unprecedented Interaction Scaling**: We reiterate the unique value proposition that distinguishes this work:

    * **Interaction Scaling**: We are the first to demonstrate an agent scaling to 2048 interactions within a fixed 40k context window with continuous performance gains (Accuracy: 3.5% $\rightarrow$ 42.5%). Mono-contextual methods structurally fail at this scale.
    * **SOTA Performance & Generalization**: IterResearch outperforms open-source agents by +14.5pp on average and matches proprietary systems (e.g., OpenAI Deep Research) on challenging benchmarks like HLE. Furthermore, **as a training-free prompting strategy**, it improves frontier models (e.g., o3, DeepSeek-V3.1) by up to 19.2pp over ReAct, proving the paradigm's effectiveness is model-agnostic.

We have comprehensively addressed all distinct concerns raised in the reviews, including providing the specific ablation studies requested during the discussion period. We respectfully believe that the extensive revisions and new experimental evidence comprehensively address the reviewers' concerns and strengthen the merit of our work.

Best regards,

Authors of Submission 15726

---

### Meta-Review · Area_Chair_vexq · 2026-01-03

**Summary:**

The reviewers raised questions regarding the novelty and framing of the proposed paradigm, the soundness of its RL/MDP formulation, the validity of the constant-context and long-horizon scaling claims, and the adequacy of empirical evidence and baseline comparisons. In the rebuttal, the authors substantially clarified the scope of their claims, refined the positioning of the contribution, and provided targeted ablations and new experimental results that directly address these concerns. Overall, the discussion resolved the major technical and conceptual issues, strengthening confidence in the paper's contributions and supporting an accept recommendation.

**Reviewer Concerns:**

### Addressed

**Clarification of the “Markovian” and constant-context claims**. The authors acknowledged the limitations of strict Markovian guarantees in open-ended LLM settings and reframed their contribution as an MDP-inspired architectural design that decouples interaction length from context cost. The title change and revised explanations appropriately narrow the scope of the claim.

**Source of performance gains (paradigm vs. prompt design)**. Through new controlled ablations comparing ReAct with report-style prompts against the full IterResearch paradigm, the authors demonstrated that gains cannot be attributed solely to prompt structure, addressing concerns about confounding factors.

**Positioning of the RL component (EAPO)**. The rebuttal clarified that EAPO is an engineering enabler rather than a theoretical RL contribution, resolving confusion about overclaiming novelty in the optimization method.

**Baseline coverage and empirical evidence**. Additional results, including newer model baselines (e.g., GPT-5), ReAct variants, and analyses across different discount factors $\gamma$ strengthened the empirical evaluation and addressed concerns about outdated or insufficient comparisons.

### Outstanding

**None**. I do not identify any major outstanding concerns that would affect the overall contribution.

**Reviewer Scores:**

**Reviewer reAn (initial score: 4)**. The reviewer questions the approach's novelty and RL formulation, raises concerns about the correctness and implications of the MDP and reward design, generalization beyond a single training model, potential information loss from iterative summarization, and insufficient positioning and comparison with closely related systems such as AlphaEvolve, while also requesting clearer experimental and cost details. In the rebuttal, the authors provided detailed responses addressing these concerns. Therefore, I would expect the reviewer to increase their score after the discussion.

**Reviewer wSFE (initial score: 4)**. The reviewer questioned the strength of the claimed constant-context and long-horizon scaling benefits, raising concerns about hidden memory and compute costs, potential inefficiencies, limited empirical validation of extrapolation and training horizons, the incremental novelty of the RL components, missing ablations, and outdated baseline comparisons. In the rebuttal, the authors addressed these concerns with detailed clarifications and additional results, including GPT-5, the ReAct variant baselines, and analyses across different $\gamma$ values. Therefore, I would expect the reviewer to increase their score after the discussion.

**Reviewer XKsG (initial scores: 4)**. The reviewer participated in the discussion prior to the ICLR data leak and explicitly acknowledged that the authors had addressed their concerns, stating that they would raise their score in support of accepting the paper.

---

### Decision · Program_Chairs · 2026-01-26

Accept (Poster)